# Antimicrobial Biomaterial on Sutures, Bandages and Face Masks with Potential for Infection Control

**DOI:** 10.3390/polym14101932

**Published:** 2022-05-10

**Authors:** Zehra Edis, Samir Haj Bloukh, Hamed Abu Sara, Nur Izyan Wan Azelee

**Affiliations:** 1Department of Pharmaceutical Sciences, College of Pharmacy and Health Sciences, Ajman University, Ajman P.O. Box 346, United Arab Emirates; 2Center of Medical and Bio-Allied Health Sciences Research, Ajman University, Ajman P.O. Box 346, United Arab Emirates; s.bloukh@ajman.ac.ae (S.H.B.); h.abusara@ajman.ac.ae (H.A.S.); 3Department of Clinical Sciences, College of Pharmacy and Health Sciences, Ajman University, Ajman P.O. Box 346, United Arab Emirates; 4Institute of Bioproduct Development (IBD), Universiti Teknologi Malaysia, Skudai 81310, Malaysia; nur.izyan@utm.my

**Keywords:** COVID-19, antimicrobial resistance, cinnamic acid, *Aloe Vera*, surgical site infection, iodophors

## Abstract

Antimicrobial resistance (AMR) is a challenge for the survival of the human race. The steady rise of resistant microorganisms against the common antimicrobials results in increased morbidity and mortality rates. Iodine and a plethora of plant secondary metabolites inhibit microbial proliferation. Antiseptic iodophors and many phytochemicals are unaffected by AMR. Surgical site and wound infections can be prevented or treated by utilizing such compounds on sutures and bandages. Coating surgical face masks with these antimicrobials can reduce microbial infections and attenuate their burden on the environment by re-use. The facile combination of *Aloe Vera Barbadensis* Miller (AV), Trans-cinnamic acid (TCA) and Iodine (I_2_) encapsulated in a polyvinylpyrrolidone (PVP) matrix seems a promising alternative to common antimicrobials. The AV-PVP-TCA-I_2_ formulation was impregnated into sterile discs, medical gauze bandages, surgical sutures and face masks. Morphology, purity and composition were confirmed by several analytical methods. Antimicrobial activity of AV-PVP-TCA-I_2_ was investigated by disc diffusion methods against ten microbial strains in comparison to gentamycin and nystatin. AV-PVP-TCA-I_2_ showed excellent antifungal and strong to intermediate antibacterial activities against most of the selected pathogens, especially in bandages and face masks. The title compound has potential use for prevention or treatment of surgical site and wound infections. Coating disposable face masks with AV-PVP-TCA-I_2_ may be a sustainable solution for their re-use and waste management.

## 1. Introduction

The COVID-19 pandemic presented the most serious challenge for mankind in this century and exacerbated another “silent pandemic” of antimicrobial resistance (AMR) [1]. The severity of the COVID-19 crisis was elevated by insufficient health care support, supply shortages of personal protective equipment and existing antimicrobial resistance [1,2,3,4,5]. Resistant ESKAPE (*Enterococcus faecium*, *Staphylococcus aureus*, *Klebsiella pneumoniae*, *Acinetobacter baumannii*, *Pseudomonas aeruginosa*, *Enterobacter* spp., and *Escherichia coli*) pathogens augmented morbidity and mortality rates [4,5,6,7,8,9]. ESKAPE pathogens are increasingly resistant to common antibiotics, drugs and antimicrobials due to inadequate utilization of antimicrobial agents [1,3,4,5,6,7,8,9]. Global overuse of antimicrobials in hospital settings without proper surveillance during the COVID-19 pandemic escalated AMR dramatically [1]. Public and health sector settings are steadily encroached by such resistant, opportunistic microorganisms [1,3,4,5,6,7,8,9]. Morbidity and mortality rates among severely ill, immunocompromised patients skyrocketed due to nosocomial infections [6,9]. The outcomes include longer duration of treatment, delayed healing processes, exponentially growing cost of treatment, increased morbidity and surging mortality rates among patients [1,3,9]. Hospital- and community-acquired infections are caused by microbes lingering in the immediate environment [1,3,4,5,6,9,10,11]. Further pandemics are expected with aggravated fatality rates globally. Multiple-drug-resistant microorganisms steadily build up their defenses by adjusting to existing conventional antimicrobial agents through survival techniques [11]. Virulence factors, such as inter-kingdom biofilm formation, are examples of their “intelligent” tools to gain multidrug and AMR [11]. These methods allow them to proliferate in all settings [11]. A new generation of agents is needed to overcome AMR-related morbidity and mortality.

Iodine, a well-known microbicide, is marketed in the form of different iodophors and polymeric complexes [12,13,14,15,16,17,18,19,20,21,22,23,24,25,26,27,28]. Antiseptic iodophors are unaffected by microbial resistance mechanisms and can be utilized to overcome AMR [19]. Iodine readily forms a variety of polyiodide ions through donor–acceptor interactions between iodides and iodine molecules [29,30,31,32,33,34,35,36]. Therefore, there are many applications of polyiodides in many fields [30,31,32,33,34,35,36]. Iodine, a small, lipophilic molecule, is released and diffuses through cell membranes [18,19,36]. It acts as an oxidizing agent and denaturates enzymes, as well as proteins, through iodination [18,19,36]. However, iodine supposedly causes discoloration of skin, pain and irritation [37]. Such side effects and uncontrolled iodine release hamper its popularity among disinfectant antiseptics [37]. Triiodides are the most stable polyiodide species. Triiodides exist in the form of asymmetric or “smart” linear, symmetric units with halogen bonding character within complexes [36]. Complexed “smart” triiodides can function as slow-iodine-releasing reservoirs due to their stability [36]. These properties mitigate the adverse effects on the skin due to reduced iodine content [21,22,36]. The complexing polymer used in our investigations is polyvinylpyrrolidone (PVP). PVP is a stabilizing matrix and acts as a reservoir for I_2_ molecules in the form of PVPI in medical applications [12,16,17,18,19,20,21,22,23,24]. We previously investigated the inhibitory actions of different formulations with silver nanoparticles, plant bio-compounds, PVP and iodine against selections of microorganisms [38,39,40,41,42]. Our formulations showed excellent to intermediate antimicrobial activities on discs and biodegradable polyglycolic acid (PGA) sutures. PGA sutures are multifilamented, biocompatible and non-toxic but feasible for biofilm formation due to their extended surface area [43,44]. Conventional surgical sutures and wound bandages are used for wound closure but can lead to surgical site infections (SSI) [43,44]. Incorporating medicinal plants on sutures, bandages and wound dressing materials can address AMR-related problems [44,45]. Pathogens are susceptible to biosynthesized antimicrobials and disinfectants.

Plants contain a rich spectrum of bioactive compounds, which display excellent antimicrobial activities through synergistic mechanisms in their defense against microorganisms [45,46,47]. Such bioactive secondary plant metabolites are the reason for the use of medicinal plants, herbs and spices throughout centuries [46,47]. Phenolic acids, polyphenols, flavonoids, hydroxycinnamic acids and other compounds are used as antimicrobial agents in an increasing number of investigations [38,39,40,41,42,43,44,45,46,47,48,49,50,51,52,53,54,55,56]. Sutures, bandages and wound dressings incorporating phytochemicals are promising alternatives to conventional treatment options due to targeted, on-site drug delivery [43,44,45,46,47,48,49,50,51,52,53,54,55,56].

Hydroxycinnamic acids are phytochemicals, which exert inhibitory action on a wide range of pathogens [46,47,55,56]. Trans-cinnamic acid (TCA) is a potent inhibitor of microbial proliferation [39,46,47,55,56,57,58,59,60]. TCA is abundantly available in the plant kingdom [46,47]. *Aloe Vera Barbadensis* Miller (AV) contains TCA and further biocomponents, which display synergistic mechanisms and therefore potentiate antiproliferative action on microorganisms [39,41,42,53,54,61]. AV has been known and utilized for centuries for its health-promoting, moisturizing and healing properties [39,41,42,53,54,61]. AV constituents include aloin, aloe-emodin, acemannan, hesperidin and further phytochemicals with antioxidant, antiproliferative and anti-inflammatory properties [39,41,42,53,54,61]. These compounds exert antiviral and even anti-corona-virus activities [61,62,63,64,65,66,67,68,69,70,71]. Such properties enable their use in wound treatment on sutures, bandages and wound dressing materials, as well as surgical face masks [72,73,74,75,76,77,78]. Coating or incorporating phytochemicals into surgical face masks can reduce the demand during supply shortages. The re-use of face masks will alleviate and moderate stockpiles of dumped face masks as part of a waste management strategy [73]. Recycling mitigates the economic burden for the end user, as well as the production and waste disposal processes [73]. An increasing number of investigations on antimicrobial materials showcase different approaches for sustainable, re-usable or even biodegradable face masks [73,74,75,76,77,78,79]. Cinnamic acid and its derivatives are increasingly used to design novel antimicrobial biomaterials within polymeric matrices [80,81,82]. Biodegradable, hydrophilic materials with antifouling and wound-healing abilities support tissue repair processes [83,84,85,86,87]. AV has stabilizing and moisturizing purposes, in addition to its anti-inflammatory qualities. TCA is a widely available bioactive phytochemical in plants with antimicrobial properties. Combining phytochemicals with iodine and PVP can potentiate the antimicrobial activities [38,39,40,41,42,88]. Consequently, the staining effect of iodine can be mitigated by reducing the iodine content without losing its inhibitory action. According to Kessler et al., skin irritation and cytotoxic effects are not due to molecular iodine in commercial PVP-I_2_ disinfectants but are caused by other additives in the formulations [89]. The release pattern of iodine and polyiodides govern the stability of the products, skin discoloration and antimicrobial activities [89,90,91,92]. PVP serves, in general, as a stabilizing and encapsulating agent of iodine moieties with the aim to reduce iodine content in the formulation and allow a controlled release.

The main purpose of this work is to present a facile, one-pot synthesis of an antimicrobial agent out of well-known, non-toxic, sustainable and biocompatible components. We combined iodine, AV, TCA and PVP within a formulation. We investigated the formation of triiodide species within the title compound in comparison to our previous works. Disc-diffusion tests against 10 common reference strains in comparison to gentamycin and nystatin were carried out to verify the in vitro antimicrobial activity of AV-PVP-TCA-I_2_. The antimicrobial action on bandages, as well as surgical sutures, suggests the potential to prevent or treat SSI and wound infections. Coating surfaces, face masks, as well as other personal protective equipment would support against AMR and envision sustainable solutions for the future. Further in vivo investigations are needed to confirm the biological activities of AV-PVP-TCA-I_2_.

## 2. Materials and Methods

### 2.1. Materials

*Aloe vera* leaves (*Aloe barbadensis* Miller, AV) were collected from the botanical garden of Ajman University campus in December, Ajman, UAE. McFarland standard sets, disposable sterilized Petri dishes with Mueller Hinton II agar, gentamicin (9125, 30 µg/disc) and nystatin (9078, 100 IU/disc) were obtained from Liofilchem Diagnostici (Roseto degli Abruzzi (TE), Italy). *E. coli* WDCM 00013 Vitroids, *P. aeruginosa* WDCM 00026 Vitroids, *K. pneumoniae* WDCM 00097 Vitroids, *C. albicans* WDCM 00054 Vitroids and *Bacillus subtilis* WDCM 0003 Vitroids were acquired from Sigma-Aldrich Chemical Co. (St. Louis, MO, USA). The reference strains *S. pneumoniae* ATCC 49619, *S. aureus* ATCC 25923, *E. faecalis* ATCC 29212, *S. pyogenes* ATCC 19615 and *P. mirabilis* ATCC 29906 were procured from Liofilchem (Roseto degli Abruzzi (TE), Italy). Polyvinylpyrrolidone (PVP-K-30), iodine (≥99.0%), Sabouraud Dextrose broth, Mueller Hinton Broth (MHB) and ethanol (analytical grade) were purchased from Sigma Aldrich (St. Louis, MO, USA). Sterile filter paper discs (diameter of 6 mm) were obtained from Himedia (Jaitala Nagpur, Maharashtra, India). Sterile polyglycolic acid (PGA) surgical sutures (DAMACRYL, 75 cm, USP: 3-0, Metric:2, 19 mm, DC3K19) were received from General Medical Disposable (GMD), GMD Group A.S., Istanbul, Turkey. Bandages and surgical, disposable, 3-ply non-woven face masks (FOMED, Qianjiang, China) were purchased from the local pharmacy. In all experiments, absolute ethanol and ultrapure water were used. All reagents were of analytical grade and were employed as delivered.

### 2.2. Preparation of Aloe vera (AV) Extract

We were kindly provided with leaves of an *Aloe vera* (*Aloe barbadensis* Miller) plant from the botanical garden of Ajman University at beginning of December between 8 and 9 am. The leaves were immediately taken to the research laboratory of the College of Pharmacy and Health Sciences. The leaves with a size between 35 to 50 cm were cleaned by tissue and then washed with water to remove dust and soil. Afterward, the clean leaves were rinsed several times with distilled water, pure ethanol, ultrapure water and dried carefully. The AV leaves were cut with a sterile knife to scrap the mucilaginous gel out. This pure gel was mixed at maximum speed for 20 min and centrifuged at 4000 rpm for 40 min (3K 30; Sigma Laborzentrifugen GmbH, Osterode am Harz, Germany). The supernatant with a light-yellow color was filled immediately into a brown bottle with a screw cap and stored in darkness at 3 °C.

### 2.3. Preparation of AV-PVP-TCA and AV-PVP-TCA-I_2_

The stock solution AV-PVP is prepared by adding 2 mL pure AV gel into 2 mL of a solution of 1 g polyvinylpyrrolidone K-30 (PVP) in 10 mL distilled water under continuous stirring at room temperature (RT). For the preparation of AV-PVP-TCA, first 0.148 g TCA is dissolved in 10 mL ethanol. Then, 2 mL of this solution is added to AV-PVP under continuous stirring at RT. After that, iodine solution is obtained by dissolving 0.05 g of iodine in 3 mL ethanol in a covered beaker at RT under stirring. An amount of 1 mL of this iodine solution is added to AV-PVP-TCA under continuous stirring at RT for the preparation of AV-PVP-TCA-I_2_.

### 2.4. Characterization of AV Complexes

The title compounds were analyzed by SEM/EDS, Raman Spectroscopy, UV-vis, FTIR and X-ray diffraction (XRD). These investigations confirmed the composition of our biomaterials.

#### 2.4.1. Scanning Electron Microscopy (SEM) and Energy-Dispersive X-ray Spectroscopy (EDX)

The scanning electron microscopy (SEM) and the energy-dispersive X-ray spectroscopy (EDS) analyses were performed with VEGA3 from Tescan (Brno, Czech Republic) at 15 kV by. AV-PVP-TCA-I_2_ was diluted by adding one drop into distilled water and positioning it on a carbon-coated copper grid. After drying the sample, it was coated with gold through the Quorum Technology Mini Sputter Coater. SEM and EDS analyses were used to determine the morphology and elemental composition of the sample, respectively.

#### 2.4.2. UV-Vis Spectrophotometry (UV-Vis)

The formulations AV-PVP-TCA and AV-PVP-TCA-I_2_ were analyzed by UV-vis spectrophotometry. The investigation was performed by a UV-Vis spectrophotometer model 2600i from Shimadzu (Kyoto, Japan) in the wavelength range of 195 to 800 nm.

#### 2.4.3. Raman Spectroscopy

The biohybrid AV-PVP-TCA-I_2_ underwent Raman analysis on a RENISHAW (Gloucestershire, UK) equipped with an optical microscope room at RT. The sample was placed into a cuvette (1 cm × 1 cm) and put in front of the laser beam. The solid-state laser beam had an excitation of 785 nm and was directed onto the sample by the 50× objective lens of a confocal microscope with a spot diameter of 2 microns. The scattered light was collected by a CCD-based monochromator with a spectral range of between 50 and 3400 cm^−1^. The spectral resolution was −1 cm^−1^, the output power was 0.5%, and the integration time was −30 s.

#### 2.4.4. Fourier Transform Infrared (FTIR) Spectroscopy

The FTIR analysis of the formulations AV-PVP-TCA and AV-PVP-TCA-I_2_ was conducted on an ATR IR spectrometer equipped with a Diamond window (Shimadzu, Kyoto, Japan). Both formulations were freeze dried and analyzed in the range between 400 to 4000 cm^−1^.

#### 2.4.5. X-ray Diffraction (XRD)

The X-ray diffraction analysis was performed by a XRD from BRUKER (D8 Advance, Karlsruhe, Germany). The formulation AV-PVP-TCA-I_2_ was analyzed by Cu radiation with a wavelength of 1.54060, coupled Two Theta/Theta, time/step of 0.5 s and a step size of 0.03.

### 2.5. Bacterial Strains and Culturing

The antimicrobial testing was performed with the reference microbial strains of *S. pneumoniae* ATCC 49619, *S. aureus* ATCC 25923, *E. faecalis* ATCC 29212, *S. pyogenes* ATCC 19615, *Bacillus subtilis* WDCM 0003 Vitroids, *P. mirabilis* ATCC 29906, *E. coli* WDCM 00013 Vitroids, *P. aeruginosa* WDCM 00026 Vitroids, *K. pneumoniae* WDCM 00097 Vitroids and *C. albicans* WDCM 00054 Vitroids. These reference strains were kept at −20 °C. The inoculation was performed by adding the fresh microbes to MHB. These suspensions were kept at 4 °C until needed.

### 2.6. Determination of Antimicrobial Properties of AV-PVP-TCA-I_2_

The inhibitory action of AV-PVP-TCA-I_2_ against nine reference bacterial strains (*S. pneumoniae* ATCC 49619, *S. aureus* ATCC 25923, *S. pyogenes* ATCC 19615, *E. faecalis* ATCC 29212, and *B. subtilis* WDCM 00003, *P. mirabilis* ATCC 29906, *P. aeruginosa* WDCM 00026, *E. coli* WDCM 00013 and *K. pneumoniae* WDCM 00097) was compared to the antibiotic gentamicin (positive control). The antifungal activity of the title compound was tested on *C. albicans* WDCM 00054 in comparison to the antibiotic nystatin (positive control). The negative controls of ethanol and water showed no susceptibility and were not mentioned further. The antimicrobial tests on discs, bandages, surgical face masks and sutures were repeated three times. The average of the independent experiments was presented in this investigation.

#### 2.6.1. Procedure for Zone of Inhibition Plate Studies

The zone of inhibition plate method was used to investigate the susceptibility of the selected pathogens toward AV-PVP-TCA-I_2_ [93]. We suspended the selected bacterial strains in 10 mL MHB and incubated at 37 °C for 2 to 4 h. *C. albicans* WDCM 00054 was incubated on Sabouraud Dextrose broth at 30 °C. The microbial cultures were adjusted to 0.5 McFarland standard. The disposable, sterilized Petri dishes with MHA were uniformly seeded with 100 μL microbial culture with sterile cotton swabs and dried for 10 min to be used for the antimicrobial testing.

#### 2.6.2. Disc Diffusion Method

The antimicrobial testing was performed against the antibiotic discs of gentamycin and nystatin following the recommendations of the Clinical and Laboratory Standards Institute (CLSI) [94]. Sterile filter paper discs were coated for 24 h with 2 mL of AV-PVP-TCA-I_2_ (11 µg/mL, 5.5 µg/mL, 2.75, and 1.38 µg/mL). After removing the discs from the solution, we dried the discs for 24 h under ambient conditions. *C. albicans* WDCM 00054 was incubated for 24 h at 30 °C on agar plates. The diameter of the zone of inhibition (ZOI) was measured with a ruler to the nearest millimeter. The antimicrobial properties of AV-PVP-TCA-I_2_ are evident from the diameters of the clear inhibition zone around the disc. No inhibition zone confirms the resistance of the microorganisms.

### 2.7. Preparation and Analysis of Impregnated Sutures, Bandages and Surgical Face Masks

The uncoated, sterile, multifilamented surgical PGA sutures of 2.5 cm were impregnated with AV-PVP-TCA-I_2_ for 18 h into 50 mL of AV-PVP-TCA-I_2_ solution (11 µg/mL) at RT. The blue sutures became brown blue and were then dried for 24 h under ambient conditions. The bandages and surgical face masks were cut to square pieces of (5 cm × 5 cm), also being impregnated in 50 mL of our title formulation (11 µg/mL) for 18 h at RT and dried for 24 h at RT. These dip-coated, dried sutures, bandages and surgical face masks were tested in vitro by ZOI assay against our selection of 10 microbial strains (*S. pneumoniae* ATCC 49619, *S. aureus* ATCC 25923, *E. faecalis* ATCC 29212, *S. pyogenes* ATCC 19615, *Bacillus subtilis* WDCM 00003, *E. coli* WDCM 00013, *P. aeruginosa* WDCM 00026, *P. mirabilis* ATCC 29906, *K. pneumoniae* WDCM 00097 and *C. albicans* WDCM 00054).

### 2.8. Statistical Analysis

We utilized SPSS software (version 17.0, SPSS Inc., Chicago, IL, USA) in our statistical analysis. The data are represented as mean values. The statistical significance between groups is calculated by one-way ANOVA. Any value of *p* < 0.05 was considered statistically significant.

## 3. Results and Discussion

AMR is a growing concern, and it increasingly endangers the existence of mankind, in particular [1]. Worldwide, the unjustified use of antimicrobials during the COVID-19 pandemic elevated the AMR crisis [1]. ESKAPE pathogens develop resistance against common antimicrobials, exacerbating the morbidity and mortality rates [1,3,4,5,6,7,8,9]. Nosocomial infections in hospital settings aggravated the suffering and fatality during the recent COVID-19 pandemic [1,2,3,5,6]. Even community-acquired infections caused by multi-drug-resistant microorganisms impacted global health due to antibiotic overuse, increasing costs and durations of treatments [1,3,4,9]. The recent COVID-19 pandemic highlighted the need for strategic solutions to overcome AMR. Confusion, misinformation, lockdowns, disruption of transport chains and supply shortages were some of the early markers of the pandemic [3]. Resource management and alternative solutions to the shortages are eminent in times of crisis and beyond [3].

The availability of antimicrobials and personal protective equipment can be a means of survival. Disinfectants are needed for inanimate surfaces in all indoor and outdoor settings, for contact-killing surfaces and as disinfectants on skin [3,11]. Preservatives are used in different health care, pharmaceutical or cosmetic products to mitigate microbial contamination. Incorporating antimicrobials on bandages, wound dressing materials and surgical sutures leads to effective prevention and treatment of SSI, as well as wound infections [11,48,49,50,51,52,53,54,55].

Health authorities enforced the use of face masks to mitigate viral transmission and droplet movement [73,74,75,76,77,78]. However, the growing number of disposed masks presents a serious source of waste and pollution [72,73]. The COVID-19 public health measures are going to be lifted as soon as the number of cases declines globally. The recommendations of social distancing, personal hygiene and wearing of face masks must be carried on in the future during known seasonal flu outbreaks. The incorporation of inhibitory biomaterials on surgical face masks can aid the prevention and treatment of upper respiratory tract infections as well [48,49,50,51,52,53,54,55]. The antiviral, anti-coronavirus activities of plant secondary metabolites are reported in previous studies and include AV, as well as several polyphenols and hydroxycinnamic acids [67,68,69,70,71]. These properties can be utilized to elevate the antiviral barrier function of face masks and their re-use. Spraying or coating face masks with antimicrobial, natural biomaterials is a sustainable solution for the planet [73,74,75,76,77,78]. Antimicrobial, non-toxic agents can be utilized to disinfect and re-use masks [73,74,75,76,77,78]. At the same time, it presents a solution for low-income populations and households under economic strain. Furthermore, the re-use of face masks can effectively alleviate the supply shortage problems in different areas globally. Antimicrobial application on face mask materials can ameliorate viral transmission processes more effectively for a longer time.

Antimicrobial agents must be sustainable and impervious to microbial resistance mechanisms. AMR is a global concern and needs to be addressed by the development of new antimicrobial agents. Plant phytochemicals are suitable candidates in the AMR challenge [44,45]. Phytochemicals and their synergistic mechanisms have assisted the survival of plants against microbes since their existence [44,45]. The antimicrobial, anti-inflammatory activities of plant constituents, such as polyphenols and hydroxycinnamic acids, have been utilized since the history of mankind [46,47].

Our title compound AV-PVP-TCA-I_2_ fits into the category of antimicrobial plant biomaterials and consists of TCA, PVP, AV, as well as iodine [39,40,41,42,95,96,97,98,99]. We used iodine, TCA and AV in our study to investigate the antimicrobial action against 10 selected pathogens. Iodine is a iodophor, which is most stable in form of “smart” triiodide species [34,36,41,42,100,101,102,103,104]. TCA, its derivatives and related AV bioconstituents, especially cinnamic acid, aloin, acemannan, aloe-emodin, pyrogallol, hesperidine, aloesin, 10-O-β-d-glucopyranosyl-aloenin and rhein increased biological activities due to their synergistic mechanisms [36,39,41,42,53,54,55,56,57,58,59,60,61,62,63,64,65,66,98,99]. TCA and its hydroxycinnamic derivates in AV have abilities to permeate cell membranes of Gram-positive pathogens, resulting in cell membrane disruptions and changes [46]. The -COOH group within TCA and AV polyphenols is weakly acidic [46,98]. Therefore, these compounds diffuse easily through the cell membranes and increase pH levels in the cytoplasm, leading to cell death [46,47]. The secondary and tertiary structures of proteins are unfolded due to the formation of hydrogen bonding. This can occur through the interaction between hydroxyl, carbonyl and carboxylate groups within TCA and AV components with the functional groups of proteins, outer membrane and cell wall components of the microorganisms [46]. The tertiary structure is unfolded by interactions of carboxylate groups by ion-ion–electrostatic interactions, resulting in disturbances in the salt bridges. Phenolic groups in TCA and the AV biocomponents interfere in the hydrophobic interactions between close, nonpolar phenyl and alkyl groups [46,47]. The covalent disulfide S-S linkages between the amino acids responsible for folding the tertiary structure of proteins are reduced to S-H groups by plant biomaterials in AV [12]. AV plant bio-compounds are oxidized by transforming their hydroxyl units to carbonyl groups [46,47]. We aimed to employ in our investigations sustainable, natural, plant-based solutions capable of acting as “smart” triiodide reservoirs [36,39,41,42]. Such triiodides are stable iodine moieties within the PVP matrix and are released upon contact with the microbial units. AV-PVP-TCA-I_2_ formulations can be promising antimicrobial biomaterials.

The analytical results of AV-PVP-TCA-I_2_ confirmed the antimicrobial properties, the formulation characteristics and sample constituents.

### 3.1. Elemental Composition and Morphological Examination of AV-PVP-TCA-I_2_

Electron Microscope (SEM) and Energy-Dispersive X-ray Spectroscopic (EDS) Analyses.

The morphology and composition of AV-PVP-TCA-I_2_ was investigated by SEM and EDS analyses (Figure 1).

The title compound AV-PVP-TCA-I_2_ reveals an amorphous morphology with interesting thread- or barrel-like forms surrounded by semi-crystalline, white patches (Figure 1a). The EDS shows carbon as the main component with 81%, followed by oxygen (6.7%) and iodine (2.5%) (Figure 1b). Chlorine, potassium and copper are originating from the AV biocompounds. Aluminum and gold appear due to their use during the preparation of the samples for the SEM analysis. The samples were coated with gold.

We investigated the antimicrobial activities of AV-PVP-TCA-I_2_-impregnated medical sutures, bandages and face masks. Sterile, braided surgical PGA sutures were dip coated with our formulations and analyzed by SEM/EDS techniques (Figure 2).

Figure 2a depicts the same plain PGA suture from our previous investigations [31]. Impregnating the suture with AV-PVP-Sage-I_2_ (11 µg/mL) results in a fully coated, homogenous surface (Figure 2b). Crystalline depositions are distributed throughout the surface of the braided suture (Figure 2b). The EDS of the impregnated suture shows only the expected composition of carbon (61%), oxygen (30.8%), potassium (3.9%), chlorine (3.2%) and iodine (1.1%) (Figure 2c). This result proves the coating process and enables these sutures for their possible use in the prevention of surgical site infections.

Plain medical face masks with and without AV-PVP-TCA-I_2_ (11 µg/mL) coating are depicted in Figure 3.

Figure 3b reveals depositions on the surface of the surgical masks. The EDS of the impregnated mask material reveals the expected compositions of carbon (82.6%), oxygen (9.8%), iodine (6,4%), calcium (0.7%) and potassium (0.5%) (Figure 3c). The plain mask depicts some rough areas, while the dip-coated mask has a smoother surface and has small aggregations on few fibers (Figure 3). These depositions formed during the drying process of the coating agent. Otherwise, there is no change in the mask material related to porosity, shape and arrangement of the fibers (Figure 3b). Such small changes will not aggravate the filtering and breathability properties of the face mask. The results may enable the use of our title material as a coating agent for face masks.

Medical bandage material was impregnated with AV-PVP-TCA-I_2_ (11 µg/mL) and analyzed by SEM/EDS techniques (Figure 4, Appendix A).

The bandage material showed two different patterns in area1 (coiled) and area 2 (ordered) in the EDS (Figure 4, Appendix A). The bandage was uniformly coated with the title compound and showed the expected composition of elements (Figure 4). The coiled area of the bandage adsorbed more iodine than the ordered part, with 35% and 6.9%, respectively (Figure 4). Again, carbon was the most abundant element, with 51.7 and 69% in the coiled and ordered structures of the bandage material. The SEM of the dip-coated bandage also shows small depositions due to the drying process, but no morphological changes to the fibers (Appendix A). The results encourage the use of the impregnated bandage for the prevention and possible treatment of wound infections.

The homogenous coating of the face mask material, sutures and bandages enables its potential use as an antimicrobial coating agent on these materials (Figure 2 and Figure 3, Appendix A). Further investigations about the stability of the coated surfaces and the effective duration of inhibitory action are needed to judge the future applications of our title compound.

The EDS verifies the purity and composition of our title formulation AV-PVP-TCA-I_2_ (Figure 1, Figure 2, Figure 3 and Figure 4). Chlorine, potassium and copper were also present in our previous study with AV-PVP-I_2_ and originate from the AV biocomponents [41].

The iodine moieties were available in every EDS of the analyzed samples (Figure 1, Figure 2, Figure 3 and Figure 4, Appendix A).

### 3.2. Spectroscopical Characterization

#### 3.2.1. Raman Spectroscopy

Raman spectroscopic analysis of AV-PVP-TCA-I_2_ is shown in Figure 5.

The Raman spectrum of the title compound AV-PVP-TCA-I_2_ shows two broad, high-intensity absorptions around 50–130 and intermediate shifts at 150–175 cm^−1^. A broad medium-sized absorption band is available from 200 to 455 cm^−1^ (Figure 5). Small-sized absorptions are available between 130 and 150 cm^−1^ (Figure 5). Such bands originate from a mixture of iodine moieties. The Raman spectrum is clearly dominated by the polyiodide absorption bands (Figure 5, Appendix A). These consist of iodide ions and molecular I_2_ within unsymmetrical polyiodide ions, as well as triiodide ions and, probably, pentaiodide ions (Figure 5, Table 1).

The Raman spectrum verifies the presence of linear, “smart” triiodide ions as a major component in the AV-PVP-TCA-I_2_ formulation, in agreement with our previous investigations (Figure 5, Table 1) [36,39,41,42].

Symmetrical, “smart” triiodide ions (I-I-I^−^) are represented by strong Raman shifts around 91 cm^−1^, followed by 100 and 112 cm^−1^ originating from symmetrical vibrations ν_1s_ (Figure 5, Table 1). Symmetrical triiodides appear at Raman shifts around 100–115 cm^−1^. According to Yushina et al., triiodides appear at 100–120 cm^−1^, bound I_2_ at 140–180 cm^−1^ and pentaiodides at 140–160 cm^−1^ [100]. The absorption mode at 112 cm^−1^ appears in our previous works of AV-PVP-Sage-I_2_ at 110 cm^−1^ [42]. The slight increase from 110 to 112 cm^−1^ confirms a blue shift with stronger bonds in the title formulation with TCA compared to the Sage biohybrid due to noncovalent interactions and the molecular surroundings of the compounds (Figure 5, Table 1) [42,100].

However, the Raman spectrum of AV-PVP-TCA-I_2_ shows further medium to very weak absorption peaks related to asymmetrical triiodide ions (I-I^…^I^−^). Savastano et al. and Xu et al. report strong symmetric stretching modes related to slightly nonlinear, symmetrical linear triiodide ions at 110 and 111 cm^−1^, respectively [23,30]. Weak to very weak absorption signals at 145 and 152 cm^−1^ confirm asymmetric vibrations, respectively. Accordingly, a medium-sized peak at 144 cm^−1^ was also available in our previous investigation of AV-PVP-Sage-I_2_ (Figure 5, Table 1) [42]. The peak at 152 cm^−1^ was accompanied by weak to very weak asymmetric stretching modes ν_as_ at 222 and 334 cm^−1^, respectively [42]. The I_3_^−^ bands have overtones in the form of increasingly weak vibrational stretching modes at 238 and 325 cm^−1^ (Figure 5, Table 1). The weak symmetric stretching mode at 221 cm^−1^ supports the linear structure, according to Savastano et al. [30]. The same Raman shift is available within the broad band in our formulations with TCA and Sage [42]. Ordinartsev et al. assign Raman modes at 116 and 235 cm^−1^ to centrosymmetric, symmetrical triiodide ions (I-I-I) [15]. Our compound shows similar symmetric stretching modes confirming the linear structure of the triiodide ions (Figure 5, Table 1) [15]. The TCA compound reveals a very low absorption intensity compared to the Sage compound, which indicates that the triiodide ions are, in the majority, linear and symmetric rather than nonlinear [42]. However, both compounds also contain slightly nonlinear, symmetrical triiodides accompanied by asymmetric triiodide ions. The title compound presents a broad, weak band between 200 and 400 cm^−1^ (Figure 5, Table 1).

Unsymmetrical triiodide ions are Raman active because they are slightly nonlinear and show absorption modes at 60, 85 and 160 cm^−1^ [30]. The same absorption bands are available in the title compound with strong- to medium-sized Raman shifts at 62, 84 and 160 cm^−1^ with lower intensity in comparison to the Sage formulation (Figure 5, Table 1) [30,42]. The strong Raman shift at 62 cm^−1^ is assigned to a hot band transition related to the ν_2_ symmetric stretching (Figure 5, Table 1) [30].

Further shoulders at 125 (weak) and 162 cm^−1^ (intermediate) verify unsymmetrical triiodide ions (Figure 5, Table 1). In particular, the Raman shifts at 62, 66, 84, 159, 166 and 169 cm^−1^ confirm the presence of distorted, nonlinear triiodide units (I-I^…^I^−^) (Figure 5, Table 1). These absorptions indicate strong I-I bonds within the unsymmetrical triiodide ions due to the noncovalent interactions. Such unsymmetrical triiodides can be simplified as (I_2_^…^I^−^). The presence of molecular iodine within such triiodide units is verified by the absorption peaks at 84, 159, 166, 169, 176, 179, 183 and 187 cm^−1^ (Figure 5, Table 1). The medium-sized shoulder at 84 cm^−1^ belongs to stretching vibrations in I_2_^…^I^−^ moieties [30]. The I_2_-unit within I^…^I^…^I^−^ is confirmed by the medium shoulders at 168 and 170 cm^−1^ (Figure 5, Table 1) [30,42].

Apart from triiodide units, characteristic absorption bands for pentaiodide ions could be assigned to 148, 157 and 165 cm^−1^ (Figure 5, Table 1). Pentaiodide ions consist of [I_2_^…^I_3_^−^] units and, therefore, are detected related to molecular iodine and triiodide ions. The availability of I_5_^−^-units seems to be manifested by the high absorption intensities and broadness of these bands between 140 and 175 cm^−1^ (Figure 5, Table 1). Nevertheless, these bands are expected to be the overtones of the triiodide ions. Pentaiodide ion absorptions are usually detected around 137–147 cm^−1^ and 167 cm^−1^ [25,82,101]. The weakness of the absorption intensities is another indicator for the absence of pentaiodide ions. At the same time, free iodine is represented by a medium-sized shoulder at 171 cm^−1^ and further weak to very weak symmetric stretching modes up to 176 cm^−1^ in AV-PVP-TCA-I_2_ (Figure 5, Table 1) [42,102,103,104]. The previous investigation with AV-PVP-Sage-I_2_ was devoid of absorption modes around 172 cm^−1^ and free iodine molecules [42,102,103,104].

The strong presence of the iodine moieties overshadows further signals in the Raman spectrum and impedes full characterization (Figure 5, Appendix A). AV biocompounds and TCA show different weak absorptions at higher Raman shifts in accordance to previous investigations [59,105,106,107]. Weak, broad bands originating from hydroxyl groups are available around the broad band between 3175 and 3400 cm^−1^ (Appendix A). The weak bands around 3045 and 3070 cm^−1^ are due to aliphatic and aromatic, unsaturated C-H stretching modes, respectively (Appendix A) [59]. The Raman spectrum is also overshadowed by the strong absorption intensities of TCA from 1050 to 2050 cm^−1^ in AV-PVP-TCA-I_2_ (Appendix A). Aromatic and aliphatic -C=C-group stretching modes are available in TCA and the title compound AV-PVP-TCA-I_2_ around 1603 and 1645 cm^−1^ in the form of strong, sharp signals, respectively (Appendix A) [59]. Both Raman shifts have elevated intensity in AV-PVP-TCA-I_2_ compared to TCA alone (Appendix A).

The deformation and stretching vibrations of unsaturated (C-H) are available in a broad band around 1200–1650 cm^−1^ with a maximum intensity at 1376 cm^−1^ (Appendix A) [59]. The stretching vibration (C-C_ring_) at 1376 cm^−1^ in AV-PVP-TCA-I_2_ appears in TCA with higher intensity at 1371 cm^−1^ (Appendix A) [59,105,106]. The lower intensity and blue shift in the title compound confirms the inclusion of TCA within AV-PVP-TCA-I_2_ (Appendix A). This Raman shift is assigned by de Souza et al. to the carboxylate ion [106]. The carbonyl groups -C=O within -COOH are more complexed in AV-PVP-TCA-I_2_ by the interaction with PVP, iodine and AV biomolecules. This is also verified by the related Raman shifts for TCA and AV-PVP-TCA-I_2_ at 1888 and 1848 cm^−1^, respectively (Appendix A) [59,105]. The -C=O stretching modes show a red shift toward 1848 cm^−1^ with increased intensity in AV-PVP-TCA-I_2_ (Appendix A). A sharp stretching mode at 2018 cm^−1^ is originating from C-C_6_H_5_ vibrations (Appendix A) [59]. The aromatic ring breathing mode is seen around 1002 and 1029 cm^−1^ (Appendix A) [59]. Very weak modes around 847 and 874 cm^−1^ are due to the aliphatic C-COOH vibrations (Appendix A) [59].

AV naturally contains TCA and acemannan [41]. Acemannan is indicated in the Raman spectrum of AV-PVP-TCA-I_2_ through the acetylation degree, with absorptions around 1740 cm^−1^ (Appendix A) [42]. The glycosidic O–C–O stretching vibrational modes appear around 1664 cm^−1^ [41]. In this work, the Raman shifts at 1602 and 1648 cm^−1^ in TCA and AV-PVP-TCA-I_2_ overshadow the glycosidic stretching modes of O-C-O groups within acemannan (Appendix A). The title compound presents higher absorption intensities for the same modes. This indicates a larger size of molecules compared to pure TCA and confirms the encapsulation of TCA.

According to Hanai et al., the Raman spectrum for cis-CA and TCA shows Raman shifts at 1632 and 1637 cm^−1^ related to the C=C-stretching vibration [59]. TCA is easily changed due to UV photoisomerization to cis-CA [58,59,107]. We propose the existence of a mixture between cis-CA and TCA in our title compound AV-PVP-TCA-I_2_ due to shifts at 1643 and 1645 cm^−1^ [58,59,107].

The Raman spectrum proves the purity of the sample AV-PVP-TCA-I_2_ through the absence of unrelated Raman shifts (Appendix A).

#### 3.2.2. UV-Vis Spectroscopy

The UV-vis spectrum of the two samples AV-PVP-TCA and AV-PVP-TCA-I_2_ is shown in Figure 6.

The UV-vis spectrum contains broad and high intensity absorptions in the regions around 200 to 230 nm and 240 to 310 nm (Figure 6). These broad absorption bands are the result of overlapping AV, TCA, PVP and iodine moieties [22,23,41,42,58,59,60]. The absorption of PVP-I_2_ (black curve) determines the regions related to PVP clearly in a broad band from 200 to 250 nm with a λ_max_ at 209 nm and a shoulder at 305 nm (Figure 6) [22,23,41,42]. The complexed iodine moieties in PVP-I_2_ absorb in the same broad region around 200 to 250 nm. They reveal further signals around λ_max_ = 288 nm and 356 nm (Figure 6). These absorptions are compliant with similar peaks in the UV-vis spectrum of AV-PVP-I_2_ and can be used to interpret the spectrum of AV-PVP-TCA-I_2_ (purple curve) (Figure 6, Table 2).

The UV-vis spectrum of AV-PVP-TCA-I_2_ confirms the composition of the sample. The UV-visible spectrum of AV-PVP-TCA-I_2_ reveals absorption peaks of molecular I_2_ (207 and 210 nm), iodide ions (202 nm) and triiodide ions (290 and 359 nm) (Figure 6, Table 2). Iodide ions, iodine molecules and triiodide ions are detected in accordance with previous investigations (Figure 6, Table 2) [15,22,23,25,27,30,33,34,36,41,42,89,90,91,100,101,102,103,104]. The bands at 290 and 359 nm depict a higher availability of symmetrical, linear triiodide ions (I-I-I^−^) compared to asymmetric triiodide ions and iodine. The predominance of “smart” triiodide ions was confirmed by the Raman spectrum (Figure 5, Table 1). The very weak absorption intensity at 359 nm is expected to be an overtone to triiodide ions instead of being an indicator of pentaiodide units.

The biohybrids AV-PVP-Sage-I_2_ and AV-PVP-I_2_ reveal comparable UV-vis absorptions to the title compound (Table 2) [41,42]. The iodine moieties appear in the UV-vis spectrum at similar wavelengths. The triiodides absorb in the Sage formulation at 291 and 359 nm (Table 2) [42]. The basic biohybrid AV-PVP-I_2_ has absorption peaks at 291 and 358 nm [41]. These triiodides are “smart” triiodides with [I-I-I] units consisting of pure halogen bonding [33,34,36,41,42]. “Smart” triiodides are more stable and have enhanced antimicrobial activities [33,34,36,41,42]. They release iodine molecules in a controlled manner when the complexes are deformed due to electrostatic interaction with the microbial cell membranes [33,34,36,41,42]. Hence, the PVP in the biohybrid AV-PVP-TCA-I_2_ complexes iodine moieties similar to the previous investigations [41,42]. The formula PVP-I_2_ can be used as PVP-I_3_^−^ according to previous reports [16,17,18,19,20,21,23,41,42]. After one-month storage of AV-PVP-TCA-I_2_, the resulting UV-vis spectrum did not show any difference compared to the UV-vis spectrum of the fresh sample.

Enhanced interactions between the iodine moieties, AV biocomponents, TCA molecules and the PVP are expressed by the UV-vis spectrum of the title compound. The broadening also refers to the increased hydrogen bonding within the sample AV-PVP-TCA-I_2_. In comparison to the PVP-I_2_ (black curve), the title compound shows increased absorption intensities for triiodides at 290 nm, while the band at 359 nm is decreased (Figure 6, Table 2). TCA molecules have strong absorptions in the region between 203 to 219, as well as 270 to 290 nm [58,59,60]. These overlap with the absorption bands of the iodine moieties and impede further clarifications (Figure 6, Table 2). The absorption bands of TCA are represented by strong bands at 210, 212, 278, 283 and 286 nm [58,59,60]. The addition of iodine leads to a hyperchromic effect, which coincides with increased availability of chromophores and conjugated systems. The mentioned bands increase in intensity in the title compound AV-PVP-TCA-I_2_, indicating the liberation of TCA and further AV components from the encapsulation by PVP (Figure 6, Table 2). The freed molecules perform hydrogen bonding, resulting in higher intensity and broadened bands in the spectrum of AV-PVP-TCA-I_2_ (Figure 6, Table 2).

PVP shows strong absorption at 203–219 and 305 nm (Figure 6, Table 2) [22,23,41,42]. Iodine moieties and TCA overshadow the region between 203 to 219 nm, but the shoulder at 305 nm can be observed easily in the UV-visible spectrum (Figure 6, Table 2). Starting with PVP-I_2_ (black curve), the intensity of the shoulder at 305 nm increases from AV-PVP-TCA to AV-PVP-TCA-I_2_ (Figure 6, Table 2).

The vibrational mode for –C=O can be observed in PVP-I_2_ at 221 nm and undergoes a blue shift toward 215 nm in AV-PVP-TCA (Figure 6, Table 2) [22,23,41,42]. The addition of iodine into the latter compound results in a red shift, with increased absorption intensity at 217 nm (Figure 6, Table 2). In comparison with AV-PVP-TCA, this bathochromic effect means for AV-PVP-TCA-I_2_ an increase in chromophores and conjugated systems related to –C=O, as well as less encapsulation and decreased hydrogen bonding. Adding iodine into the system liberates TCA and the AV components from the encapsulation by PVP. Their –C=O bonds absorb at higher intensities, while PVP encapsulates the iodine units.

The biohybrid, as well as AV-PVP-TCA-I_2_, shows higher absorbance between the wavelength range of 240 to 320 nm compared to our previously reported complex AV-PVP-Sage-I_2_ [42]. This hyperchromic effect reveals the existence of more π-electrons in the title compound. This means an increase in conjugation, less complexation, less encapsulation and less hydrogen bonding. TCA aromatic rings absorb around 277 nm and are, together with the AV aromatic biocomponents, the reason for the broad signal between 260 to 320 nm in the UV-vis spectrum. TCA has an absorption maximum at λ-max = 274 nm, with additional absorption peaks at 215 and 204 nm in accordance with previous investigations (Figure 6, Table 2) [58,59,60]. However, TCA overshadows the UV-vis spectrum and makes further clarifications about AV biocomponents difficult (Figure 6, Table 2). According to Saleh et al., cis-CA absorbs at 255 nm [58]. The additional shoulder at 256 nm in AV-PVP-TCA-I_2_ may be related to cis-CA as a UV-induced photoisomerization of TCA (Figure 6) [58,59,107]. Therefore, we may confirm a mixture between TCA and cis-CA in the sample after iodination. The same result was obtained during the discussion of the Raman spectra (Appendix A). The broadness of the absorption band in the UV-vis around 260 to 320 nm for both of the compounds AV-PVP-TCA and AV-PVP-TCA-I_2_ may even suggest the availability of cis-CA before iodination.

#### 3.2.3. Fourier Transform Infrared (FTIR) Spectroscopy

The FTIR analysis of AV-PVP-TCA and AV-PVP-TCA-I_2_ augments the small differences between the two samples clearly (Figure 7, Appendix A).

The title compounds AV-PVP-TCA-I_2_ and AV-PVP-TCA reveal a similar pattern in the transmission spectrum (Figure 7). The iodinated biomaterial shows intense broadening and higher absorption intensity in two regions. These are evident around 3308, 1632, 1639, 1649 and 1655 cm^−1^, being related to stretching vibrations of –COOH, –C=O (-COOH) and C=C groups, respectively (Figure 7, Table 3) [39,41,42,59,108,109,110].

The addition of iodine into AV-PVP-TCA produces shifts in the vibrational bands from 1646, 1651 and 1656 cm^−1^ toward 1632 cm^−1^, 1639 cm^−1^ and 1655 cm^−1^, respectively (Figure 7, Table 3) [59]. The blue shift from 1646 to 1632 cm^–1^ is related to the C=C bonds within TCA after iodine addition (Table 3). The C=O of TCA underwent a small shift from 1656 to 1655 cm^–1^ after iodination (Table 3). The carbonyl group in PVP, which is responsible for complexing iodine moieties, showed a blue shift from 1651 to 1639 cm^–1^ after the addition of iodine (Table 3) [19,22,59]. The blue shifts are due to the interaction of iodine with the C=O groups of PVP, verifying the encapsulation of the polyiodide ions by the PVP polymer matrix. This is also confirmed by a blue shift of the vibrational band related to PVP amide groups from 1292 to 1288 cm^–1^ after iodination (Table 3) [19]. The broadening, coupled with the increase in absorption intensities, indicates a higher availability of -C=C-, carboxyl and carbonyl groups originating from the released TCA and AV biocomponents. These molecules interact with the light and engage in hydrogen bonding. Adding molecular iodine also triggers oxidation reduction reactions. Iodine is reduced to iodide-ions, while the hydroxyl groups of AV biocomponents are oxidized to carbonyl groups. The encapsulation of PVP increases with the addition of iodine (Figure 7, Table 3).

The FTIR analysis reveals striking similarities to our previous investigations (Figure 7, Table 3) [41,42]. AV-PVP-TCA and its iodinated formulation AV-PVP-TCA-I_2_ absorb in the same wavelength regions as in our previous reports (Figure 7, Table 3) [41,42]. AV biocomponents are available in the FTIR spectrum (Figure 7). Acemannan is one of the major components in AV and can be detected by the O-C-O acetyl stretching at 1275 cm^−1^ in both FTIR spectra (Figure 7, Appendix A) [41]. The small intensity peak at 1113 cm^−1^ is originating from aloin and is reduced in intensity after adding I_2_ because it is also more encapsulated by the PVP complex [41,42]. The complexation of iodine moieties also releases other molecules previously complexed by the PVP backbone (Figure 7, Table 3) [41,42]. This finding is manifested in AV-PVP-TCA-I_2_ by the increased band intensities and broadening between 3100 and 3600 cm^−1^ (Figure 7, Table 3) [41,42]. This proves a higher interaction of the light with more released –COOH-groups, which connect after their release from the PVP carbonyl oxygen atoms with other biomolecules through hydrogen bonding. The same increase in intensity occurs for the bands of the carbonyl-C=O stretching vibration (Figure 6). The band appears in AV-PVP-TCA at 1646 cm^−1^ with lower intensity and undergoes a red shift, coupled with an increase in intensity toward 1632 cm^−1^ in the iodinated title compound. The increased intensity is proof of the release from encapsulation from PVP after adding iodine (Figure 7, Table 3) [41,42]. At the same time, this indicates the increase in –C=O groups as a result of oxidation (Figure 7, Table 3) [41,42]. Hydroxyl groups within the AV biocompounds are oxidized, while iodine is reduced into iodide ions. The red shift toward 1632 cm^−1^ underlines the process of more hydrogen bonding and increased interactions between the newly formed –C=O groups and other moieties within the sample (Figure 7, Table 3) [41,42].

As previously reported, adding iodine leads to a hypsochromic effect because of complexation of triiodide ions into the PVP matrix [16,17,18,19,20,21,23,41,42]. This complexation reduces the number of π-electrons and conjugation systems, which leads to more hydrogen bonding [41,42]. Once molecular iodine is added into the formulation, it is reduced by AV biocomponents to iodide, while the phytochemical functional groups are oxidized and partly released from the PVP matrix [39,41,42]. Triiodide ions form hydrogen bonding to carbonyl groups within PVP [16,17,18,19,20,21,23,41,42]. The released AV phytochemicals result in higher absorption intensities for –COOH, –OH, –C=O, -C=C- and –C–O in AV-PVP-TCA-I_2_ (Figure 7, Table 3) [41,42].

The FTIR spectrum reveals bands for PVP carbonyl groups at 1651 and 1639 cm^−1^ for AV-PVP-TCA and AV-PVP-TCA-I_2_, respectively (Figure 7, Table 3) [41,42]. This red shift indicates a stronger encapsulation of the carbonyl groups of PVP after addition of iodine, resulting in less hydrogen bonding with the AV biocomponents and TCA. Again, this proves a partial release of those components by replacement with iodine moieties. The same peak is observed for the Sage formulations at 1650 cm^−1^ for PVP [41,42]. The stretching vibration of -C-O appears for TCA at 1290 cm^−1^ before and at 1289 cm^−1^ with higher intensity after adding iodine. This also confirms the partial release of TCA molecules after adding iodine (Figure 7, Table 3) [41,42]. The methylene groups of PVP are usually located around 2700 to 2900 cm^−1^ for PVP [41,42,108,109]. In our title compounds, the bands are available at 2990/2989 cm^−1^ (C-H)_a_, 2953 (CH_2_) and 2839 (C-H)_s_ (Figure 7, Table 3) [41,42,108,109]. However, their intensities are reduced after iodine addition. This proves higher involvement of PVP entities in complexation in AV-PVP-TCA-I_2_, in accordance with previous investigations (Figure 7, Table 3) [41,42]. The same results can be observed for C-C and CH_2_ rocking vibrational bands at 1014 cm^−1^ (Figure 7, Table 3). The absorption intensity of this band is highly reduced in the title compound AV-PVP-TCA-I_2_, confirming the higher coiled structure of the PVP (Figure 7, Table 3). Our previously investigated compounds AV-PVP-Sage-I_2_, AV-PVP-I_2_, AV-PVP-I_2_-Na and pure PVP show the same band at 1017 cm^−1^ [41,42,108]. The red shift from 1017 to 1016/1014 cm^−1^ proves higher complexation in our title compounds AV-PVP-TCA/AV-PVP-TCA-I_2_ in comparison to pure PVP and our previous investigations [41,42,108]. The encapsulation of iodine moieties leads to a coiled structure of PVP, which reduces the absorption intensities of the backbone carbon chain of the polymer.

According to Hanai et al., cis-CA shows an absorption peak for the C=C-stretching vibrations at 1631 cm^−1^, which is comparable to our result of 1632 cm^−1^ in Figure 7 (Table 3) [59]. This can confirm the availability of cis-CA in our sample after iodination due to UV-induced photoisomerization (Figure 7, Table 3).

In general, incorporating iodine into the system partially releases TCA and AV biocompounds from the PVP matrix. TCA and AV components appear in the FTIR spectrum of the iodinated title compound with higher absorbance. These molecules show higher intensities in the spectrum through their available functional groups, chromophores and increased abilities to form hydrogen bonding. The increase in absorption intensities is manifested by the AV-PVP-TCA-I_2_ FTIR spectrum around 3308 (-O-H and -COOH), 1632 (asymmetric C=O from -COOH) and 1641 cm^–1^ (C=C) (Figure 7, Table 3). The bands originate from TCA and the main AV component aloin. Further indicators are the vibrational stretching bands around 1343 and 1314 cm^–1^ (both C-C) (Figure 7, Table 3). The C-O vibrational stretching bands at 1289, 1275, 1223, 1205, 1073 and 1064 cm^–1^ show higher absorbances compared to their counterparts in the AV-PVP-TCA compound.

As a result, iodine addition intensifies the -OH, -COOH, -C-O and -C=O interaction of de-complexed TCA and AV biocomponents. At the same time, higher encapsulation of iodine by the PVP backbone leads to lower intensity bands of -CH and -CH_2_ groups. The FTIR spectrum proves the purity of the synthesized biohybrids AV-PVP-TCA and AV-PVP-TCA-I_2_ (Figure 7, Table 3).

#### 3.2.4. X-ray Diffraction (XRD)

The XRD analysis of AV-PVP-TCA-I_2_ reveals sharp diffraction peaks originating from AV, TCA, PVP and iodine (Figure 8).

The two main peaks at 2θ = 17, 19° and smaller peaks at 11 and 15° belong to TCA, while others at 11 and 20° are due to PVP [41,95,96,97,98,99,100]. Further peaks reveal crystalline behavior at 2θ = 10, 15, 16, 19, 20, 21 and 22° [41,95,96,97,98,99,100]. The small broad bands from 2θ = 20° to 30° indicate the formation of amorphous phases after the deposition of polyiodide species into the PVP matrix (Figure 8). Iodide moieties are encapsulated into the PVP backbone and cause small but broad amorphous regions below the sharp peaks at 22.8, 23.8, 25.5, 27 and 30 (Figure 8, Table 4).

The decrease in peak intensities related to PVP and their shift toward 2Theta values of 11° and 20° confirm the higher encapsulation of polyiodides in the title compound compared to AV-PVP-I_2_ alone. The addition of TCA into the system clearly induced a higher crystallization degree, lower intensities and sharp peaks (Figure 8, Table 4).

The purity of our title compound AV-PVP-TCA-I_2_ is confirmed by the XRD analysis (Figure 8, Table 4).

Iodine and TCA are represented according to our previous studies and other investigations around 24–46 2Theta degrees [41,95,96,97]. The intermediate peaks around 23, 24, 26, 27 and 30 degrees (2Theta) are followed by very weak to weak signals at 37 and 46 degrees, respectively (Figure 8, Table 4). These signals appear in previous investigations and confirm the iodine within our sample [41,95,97]. Figure 8 and Table 4 indicate the related strong and intermediate peaks of the crystalline TCA at 2Theta degrees of 10, 23 and 25, 29, respectively (Figure 8, Table 4) [96]. These are accompanied by several other weak peaks at 2Theta 19, 20, 27, 32, 34 (Figure 8, Table 4) [96]. AV biocomponents are indicated by intermediate peaks at 2Theta degrees of 16 and 22 in AV-PVP-TCA-I_2_ (Figure 8, Table 4), which are all available in previous investigations as well [41,98,99]. The title compound AV-PVP-TCA-I_2_ has a higher crystalline character in comparison to our previous investigation of AV-PVP-I_2_, AV-PVP-NaI and AV-PVP-Sage-I_2_ [41,42]. The higher crystallinity in AV-PVP-TCA-I_2_ is due to the higher presence of TCA in the formulation compared to the other bio-hybrids [41,42]. The sample shows crystalline character in the SEM of the surgical suture (Figure 2). The same result was obtained in the XRD analysis with sharp, distinct peaks for AV, TCA, iodine and PVP (Figure 8, Table 4) [41,98,99,100].

### 3.3. Antimicrobial Activities of AV-PVP-TCA and AV-PVP-TCA-I_2_

Disc diffusion assay against 10 reference microorganisms was used to investigate the antimicrobial activities of AV-PVP-TCA and AV-PVP-TCA-I_2_. The utilized Gram-positive bacteria were *S. pneumonia* ATCC 49619, *S. aureus* ATCC 25923, *S. pyogenes* ATCC 19615, *E. faecalis* ATCC 29212 and *B. subtilis* WDCM0003. Gram-negative bacteria included the strains *E. coli* WDCM 00013 Vitroids, *P. mirabilis *ATCC 29906, *P. aeruginosa* WDCM 00026 Vitroids and *K. pneumonia* WDCM00097 Vitroids. *C. albicans* WDCM 00054 Vitroids, which were used to test the antifungal activities of our biomaterial. The antimicrobial properties of our bio-compounds were compared to the positive controls, gentamicin and nystatin. Methanol, ethanol and water were the negative controls. They showed no zone of inhibition (ZOI) and were excluded from Table 5.

Our title biohybrid AV-PVP-TCA-I_2_ shows remarkable antimicrobial properties of impregnated discs, sutures, bandages and face masks (Table 4). The tests were performed on discs at concentrations of 11, 5.5 and 2.75 µg/mL. Sutures, bandages and masks were treated with a concentration of 11 µg/mL. The highest inhibition zones were achieved, in general, against the fungal reference strain *C. albicans* WDCM 00054, followed by Gram-positive and, finally, Gram-negative bacterial strains.

*C. albicans* WDCM 00054, *S. aureus* ATCC 25923, *S. pneumoniae* ATCC 49619 and *E. coli* WDCM 00013 were more susceptible toward AV-PVP-TCA-I_2_ than their respective antibiotic positive controls (Table 3). AV-PVP-TCA-I_2_ inhibits *C. albicans* WDCM 00054 at concentrations of 11 and 5.5 µg/mL with 46 and 25 mm stronger than nystatin (Table 5, Figure 9a).

*S. aureus* ATCC 25923 (Figure 9b) and *K. pneumoniae* WDCM 00097 reveal a ZOI of 17 and 13 mm, respectively. The susceptibility toward AV-PVP-TCA-I_2_ decreases in the order of the Gram-positive pathogens *S. pyogenes* ATCC 19615 (16 mm), *E. faecalis* ATCC 29212 (15 mm), *S. pneumoniae* ATCC 49619 and *B. subtilis* WDCM 00003 (both 14 mm) (Table 5, Figure 9c). The Gram-negative reference strains follow with *P. aeruginosa* WDCM 00026 (13/12 mm), *K. pneumoniae* WDCM 00097 (13/10 mm) and *E. coli* WDCM 00013 (13/0 mm) at concentrations of 11 and 5.5 µg/mL, respectively (Table 5, Figure 9).

The disc diffusion tests revealed strong antifungal activity of AV-PVP-TCA-I_2_ against *C. albicans* WDCM 00054. The next most susceptible microorganism is the Gram-positive bacteria *S. aureus* ATCC 25932. Intermediate antibacterial activity against further Gram-positive and Gram-negative species *K. pneumoniae* WDCM 00097, *P. aeruginosa* WDCM 00026 and *E. coli* WDCM 00013 are manifested as well (Table 5, Figure 9).

The promising results in the disc diffusion studies tempted us to investigate AV-PVP-TCA-I_2_ on surgical sutures as potential preventive agents against surgical site infections. We coated braided PGA surgical sutures with the title compound. The inhibitory activity on surgical sutures was tested against the same 10 reference strains (Table 5, Figure 10).

The inhibitory zones around the sutures follow the same trends displayed by the disc diffusion studies on sterile discs. The dip-coated sutures showed extraordinary antifungal activity against *C. albicans* WDCM 00054, with an inhibition zone of 17 mm (Table 5, Figure 10). The Gram-positive pathogen, *S. aureus* ATCC 25,932 (7 mm), is followed by *S. pyogenes* ATCC 19615 (4 mm), *B. subtilis* WDCM 00003 (4 mm), *S. pneumoniae* ATCC 49619 and *E. faecalis* ATCC 29212 (both 3 mm) (Figure 10, Table 5). The two Gram-negative pathogens, *P. aeruginosa* WDCM 00026 and *K. pneumoniae* WDCM 00097, revealed inhibition zones of 2 mm against AV-PVP-TCA-I_2_ (Figure 10, Table 5).

We impregnated sterile face masks with the biomaterial AV-PVP-TCA-I_2_ to investigate the potential antimicrobial action on masks. The process can potentiate the re-use of masks during any pandemic. This will ensure safety, sustainability and waste reduction. Communities without adequate resources could solve this issue by re-using face masks after treatment with our biocompatible, non-toxic title compound.

The antimicrobial tests on face masks against the same 10 reference strains delivered the same trends. AV-PVP-TCA-I_2_ is a strong antifungal agent against *C. albicans* WDCM 00054 (54 mm) (Table 5, Figure 11).

The highest susceptibility against our title compound is shown by the Gram-positive *S. aureus* ATCC 25932 (35 mm), followed by *S. pneumoniae* ATCC 49619 (20 mm), *E. faecalis* ATCC 29212 (20 mm), *S. pyogenes* ATCC 19615 (19 mm) and *B. subtilis* WDCM 00003 (16 mm) (Figure 11, Table 5). The three Gram-negative pathogens, *K. pneumoniae* WDCM 00097, *P. aeruginosa* WDCM 00026 and *E. coli* WDCM 00013, displayed against AV-PVP-TCA-I_2_ inhibition zones of 21, 20 and 16 mm, respectively (Figure 10, Table 5).

We tested the impregnated sterile cotton bandages with AV-PVP-TCA-I_2_ and tested them against the same 10 microorganisms. Our compound can be potentially used as a strong microbial agent in the treatment of wounds and wound infections. The results show strong antifungal activities against *C. albicans* WDCM 00054, with inhibition zone of 55 mm (Table 5, Figure 12).

The Gram-positive *S. aureus* ATCC 25932 (40 mm) is followed in decreasing inhibitory action by *S. pneumoniae* ATCC 49619 and *S. pyogenes* ATCC 19615 (both 24 mm), *E. faecalis* ATCC 29212 and *B. subtilis* WDCM 00003 (both 23 mm) (Figure 11, Table 5).

The three Gram-negative bacteria, *K. pneumoniae* WDCM 00097, *E. coli* WDCM 00013 and *P. aeruginosa* WDCM 00026, revealed inhibition zones of 28, 26 and 20 mm, respectively (Figure 12, Table 5).

The highest inhibitory action was achieved by AV-PVP-TCA-I_2_ against the fungal strain *C. albicans* WDCM 00054 with a ZOI = 46 mm compared to nystatin with 16 mm (Table 5). We used the same method and concentration in our previous work with AV-PVP-Sage-I_2_ and reported an alleviated susceptibility of *C. albicans* WDCM 00054 with 52 mm [42]. The inhibitory action is similar or slightly weaker when the formulation contains TCA instead of Sage. The same result is seen in slightly weaker inhibition zones against *S. aureus* ATCC 25923 with 17, 13 and 11 mm (Table 5). In comparison, the Sage formulation showed inhibition zones of 20, 15 and 14 mm against the same pathogen [42]. The spore forming *B. subtilis* WDCM 0003 is slightly more inhibited by the title formulation with TCA in comparison to the Sage formulation (Table 5) [42]. When TCA replaces Sage in the formulation at a concentration of 11 µg/mL, the susceptibility of the Gram-positive *S. pyogenes* ATCC 19615 is slightly higher with 16 mm compared to 14 mm [42]. The Gram-negative, multi-drug-resistant *P. aeruginosa* WDCM 00026 is inhibited by AV-PVP-TCA-I_2_ with 13 and 12 mm, while it is resistant against AV-PVP-Sage-I_2_ (Table 5) [42,46]. The same pathogen shows augmented susceptibility against AV-PVP-TCA-I_2_ on dip-coated sutures as well (Table 5) [42]. All Gram-positive pathogens and *C. albicans* WDCM 00054 were more inhibited on AV-PVP-TCA-I_2_-coated sutures compared to AV-PVP-Sage-I_2_-coated ones (Table 5) [42]. *C. albicans* WDCM 00,054 was inhibited by AV-PVP-TCA-I_2_-coated suture with 17 mm, while the Sage-formulation-coated suture resulted in 15 mm (Table 5) [42]. We also soaked the bandages and surgical face masks with our formulation AV-PVP-TCA-I_2_. The impregnated cotton bandages and face masks showed high antimicrobial activity against *C. albicans* WDCM 00054, followed by the Gram-positive pathogen *S. aureus* ATCC 25932 and the Gram-negative *K. pneumoniae* WDCM 00097 (Table 5, Figure 11 and Figure 12).

In conclusion, the synergistic action of the Sage biocompounds in the biohybrid AV-PVP-Sage-I_2_ achieves slightly higher inhibition zones for *C. albicans* WDCM 00054 and *S. aureus* ATCC 25932 in comparison to AV-PVP-TCA-I_2_ at the same concentration in disc diffusion studies [42]. However, the title compound AV-PVP-TCA-I_2_ is effective against *P. aeruginosa* WDCM 00026, while the Sage formulation fails to inhibit this multi-drug-resistant pathogen on disc diffusion studies [42]. The AV-PVP-TCA-I_2_ impregnated sutures were more effective than the ones coated with AV-PVP-Sage-I_2_ against Gram-positive bacteria and *C. albicans* WDCM 00054 [42]. These results encourage the use of AV-PVP-TCA-I_2_ on sutures, bandages and face masks against *C. albicans* WDCM 00054, Gram-positive *S. aureus* ATCC 25932, Gram-negative *P. aeruginosa* WDCM 00026 and *K. pneumoniae* WDCM 00097.

The susceptibility of the tested microorganisms against our biomaterial AV-PVP-TCA-I_2_ reveals similar patterns to previously reported biomaterials AV-PVP-I_2_ and AV-PVP-Sage-I_2_ [41,42]. The highest susceptibility was showcased by *C. albicans* WDCM 00054, followed by high to intermediate inhibition of the Gram-positive pathogens (Table 5). Gram-negative microorganisms displayed the lowest inhibition zones due to their higher negatively charged, complicated outer cell membrane with porin channels. This confirms the antimicrobial action of molecular iodine, which does not go through porin channels like its hydrophilic iodide and triiodide counterparts. Iodine is released from triiodide ions, which are encapsulated by the PVP complex [41,42,46]. The release occurs through deformation of the PVP complex through dipol–dipol interactions of TCA and AV biocomponents [41,42,46]. Remarkably, the results show higher susceptibility of the multi-drug-resistant *P. aeruginosa* WDCM 00026 against AV-PVP-TCA-I_2_ in comparison to the Sage formulation (Table 5) [42,46]. The title formulation seems to be able to avoid the efflux pumps within Gram-negative, motile bacilli. *P. mirabilis* ATCC 29906 is a swarming bacillus with several flaggellae. *P. mirabilis* ATCC 29906 shows resistance against both of our formulations, AV-PVP-TCA-I_2_ and AV-PVP-Sage-I_2_ (Table 5) [42].

In general, the fungi *C. albicans* WDCM 00054 is highly susceptible, followed by Gram-positive and lastly Gram-negative bacteria. The bacterial morphology rules susceptibility of the studied microorganisms against AV-PVP-TCA-I_2_ and our previous compounds in the same patterns [41,42]. Cocci, non-motile, Gram-positive, higher agglomerated clusters (*S. aureus*), chains (*E. faecalis*, *S. pyogenes*) and pairs (*S. pneumoniae*) are more susceptible than motile, Gram-negative pathogens (Table 5) [41,42]. Fungus and Gram-positive microorganisms of our selection have less complicated outer cell membrane structures and are therefore more vulnerable toward AV-PVP-TCA-I_2_ [8,11,41,42,44,47,110,111,112]. The thick peptidoglycan layers of Gram-positive pathogens are crosslinked with peptide bridges, which possibly form hydrogen bonding, dipol–dipol and electrostatic interactions with components in our title formulation through TCA, AV and PVP [8,11,36,39,41,42,44,47,110,111,112]. These interactions deform the complex AV-PVP-TCA-I_2_ and result in the release of molecular iodine from the “smart” triiodides reservoir [36,39,41,42]. Iodine compromises the cell membranes, leads to cell leakage, diffuses easily through the cell membranes and causes cell death by inhibiting important cell functions [17,18,36,39,41,42].

In conclusion, the biomaterial is a strong antifungal agent on bandages, discs, sutures and masks against *C. albicans* WDCM 00054. All the tested materials exhibited highest inhibitory action against the Gram-positive pathogen *S. aureus* ATCC 25932 and other tested bacterial strains. Intermediate to high inhibitory action is recorded with the Gram-negative pathogens, *K. pneumoniae* WDCM 00097, *E. coli* WDCM 00013 and *P. aeruginosa* WDCM 00026. *P. mirabilis* ATCC 29906 showed resistance in all the investigations.

Our title compound AV-PVP-TCA-I_2_ is a combination of affordable, commonly used and well-known non-toxic antimicrobial agents. Coating common gauze bandages and surgical sutures with AV-PVP-TCA-I_2_ can mitigate or even prevent surgical site and wound infections. The work is based on our previous investigations [36,39,41,42]. Impregnating face masks with AV-PVP-TCA-I_2_ can potentially reduce the viral load in the area between the mask and the face during its use. AV-PVP-TCA-I_2_ can deactivate common microorganisms, moisturize skin and tissues and alleviate the filtering properties of the mask fibers. The formulation AV-PVP-TCA-I_2_ may potentially support the community during pandemics, crises, shortages of antimicrobial disinfectants and reduce the need for disposing face masks after single use. The components are readily available for any individual and can be produced in households. Further investigations are needed to confirm the suggested uses and other biological activities of the formulation.

## 4. Conclusions

Iodine-based disinfectants and antiseptics are impervious to AMR and can play a strategic role against resistant, opportunistic pathogens. Biomaterials consisting of iodine and plant biocompounds have enhanced inhibitory action against a variety of microorganisms. These microbicidal formulations can be employed as disinfectants on inanimate surfaces to mitigate community- and hospital-acquired infections, as well as antiseptics on skin tissues. Coating sutures, wound dressing materials and gauze bandages can alleviate the development of SSI and wound infections. Surgical face masks are needed in the prevention of microbial infections through air droplets. Single-use, disposable face masks already place a huge burden on the environment, resources and low-income populations. Impregnating face masks with antimicrobial formulations can alleviate viral transmission and ameliorate sustainability. The title compound AV-PVP-TCA-I_2_ showed excellent antifungal and good to intermediate antibacterial activities against our selection of Gram-positive- and Gram-negative reference strains, respectively. AV-PVP-TCA-I_2_ inhibits *C. albicans* WDCM 00054 remarkably on sutures, bandages and face masks with a ZOI of 17, 55 and 54 mm, respectively. Gram-positive *S. aureus* ATCC 25932, Gram-negative *P. aeruginosa* WDCM 00026 and *K. pneumoniae* WDCM 00097 are also strongly susceptible on bandages and face masks. Therefore, the title compound has the potential to be used as an infection control agent against the mentioned pathogens. The composition and morphology of the title compound was verified by analytical methods. Our results confirm AV-PVP-TCA-I_2_ as a strong antifungal agent against *C. albicans*. The title compound is an alternative in the prevention or treatment of infectious diseases on inanimate surfaces, face masks, sutures and bandages. Further investigations are needed to confirm the intended uses and other biological activities.

## Figures and Tables

**Figure 1 polymers-14-01932-f001:**
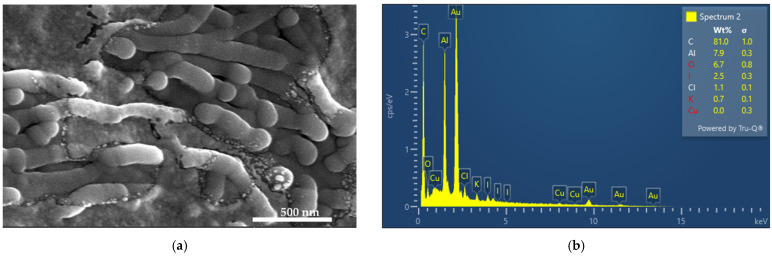
SEM (**a**) and EDS (**b**) of AV-PVP-TCA-I_2_.

**Figure 2 polymers-14-01932-f002:**
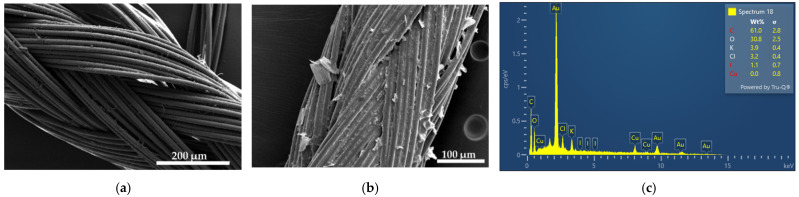
SEM of sutures: (**a**) plain suture [31]; (**b**) impregnated with AV-PVP-TCA-I_2_; (**c**) EDS of coated sutures.

**Figure 3 polymers-14-01932-f003:**
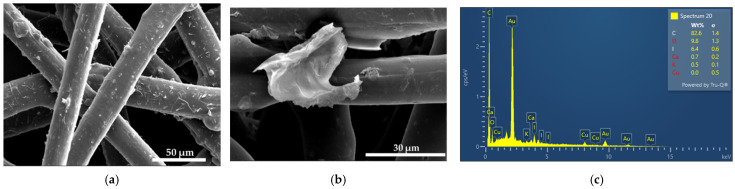
SEM of surgical face masks: (**a**) plain mask; (**b**) dip coated with AV-PVP-TCA-I_2_; (**c**) EDS of coated face mask.

**Figure 4 polymers-14-01932-f004:**
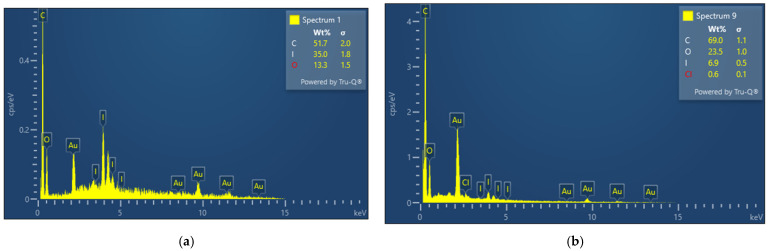
EDS of medical bandages impregnated with AV-PVP-TCA-I_2_: (**a**) area 1 coiled; (**b**) area 2 ordered.

**Figure 5 polymers-14-01932-f005:**
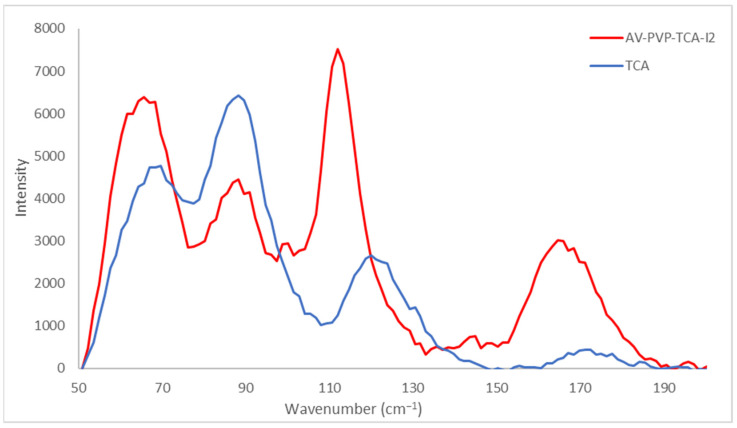
Raman spectroscopic analysis of AV-PVP-TCA-I_2_ and TCA.

**Figure 6 polymers-14-01932-f006:**
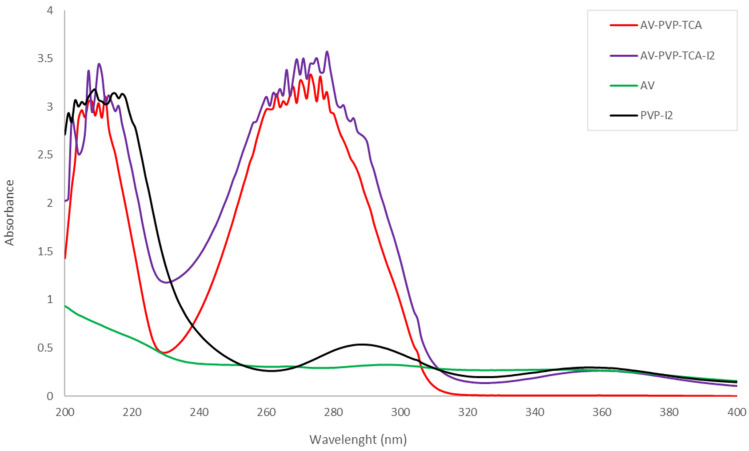
UV-vis analysis of AV-PVP-TCA, AV-PVP-TCA-I_2_, AV and PVP-I_2_ (200–400 nm). (AV-PVP-TCA: red; AV-PVP-TCA-I_2_: purple; AV: green; PVP- I_2_: black).

**Figure 7 polymers-14-01932-f007:**
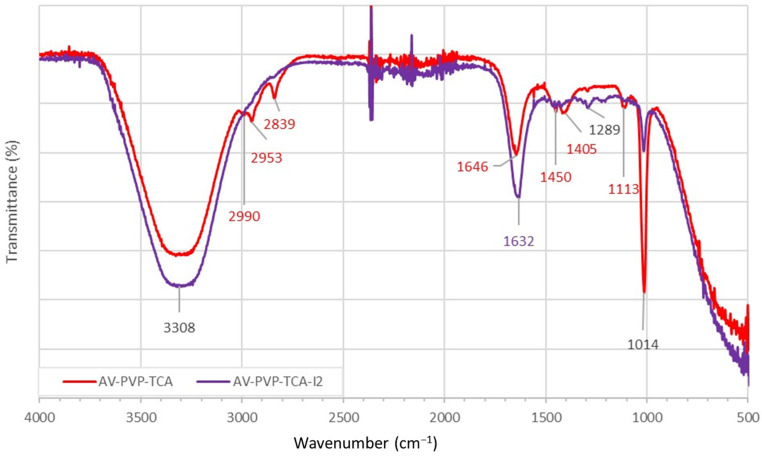
FTIR spectroscopic analysis from AV-PVP-TCA (red) and AV-PVP-TCA-I_2_ (purple).

**Figure 8 polymers-14-01932-f008:**
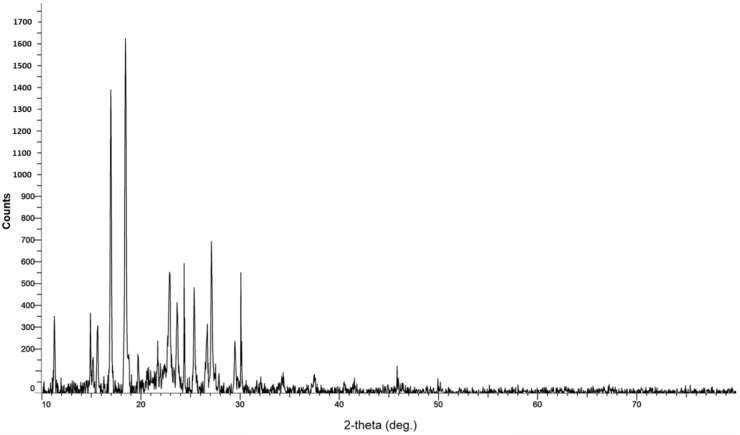
XRD analysis of AV-PVP-TCA-I_2_.

**Figure 9 polymers-14-01932-f009:**
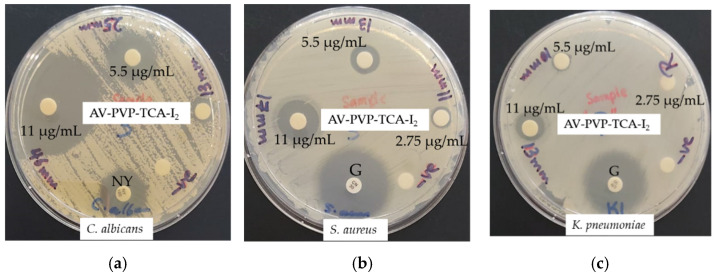
Disc diffusion assay of AV-PVP-TCA-I_2_ (with concentrations of 11, 5.5 and 2.75 µg/mL) with positive control antibiotic nystatin (NY = 100 IU) and gentamicin (G = 30 µg/disc). From left to right: AV-PVP-TCA-I_2_ against (**a**) *C. albicans* WDCM 00054; (**b**) *S. aureus* ATCC 25932; (**c**) *K. pneumoniae* WDCM 00097.

**Figure 10 polymers-14-01932-f010:**
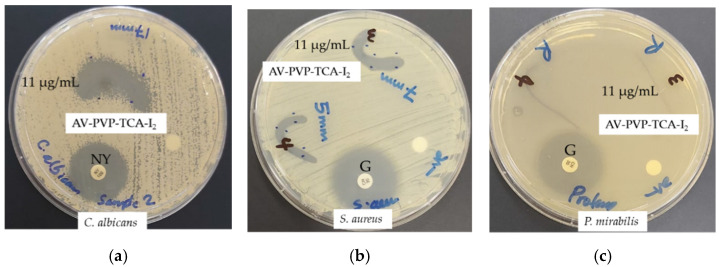
Impregnated, sterile PGA sutures with AV-PVP-TCA-I_2_. with positive control antibiotic nystatin (NY = 100 IU) and gentamicin (G = 30 µg/disc). From left to right: AV-PVP-TCA-I_2_ (11 µg/mL) against (**a**) *C. albicans* WDCM 00054; (**b**) *S. aureus* ATCC 25932; (**c**) *P. mirabilis* ATCC 29906.

**Figure 11 polymers-14-01932-f011:**
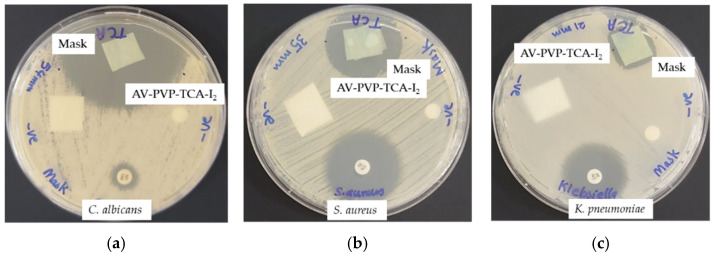
Impregnated sterile masks with AV-PVP-TCA-I_2_. with positive control antibiotic nystatin (NY = 100 IU) and gentamicin (G = 30 µg/disc). From left to right: AV-PVP-TCA-I_2_ (11 µg/mL) against (**a**) *C. albicans* WDCM 00054; (**b**) *S. aureus* ATCC 25932; (**c**) *K. pneumoniae* WDCM 00097.

**Figure 12 polymers-14-01932-f012:**
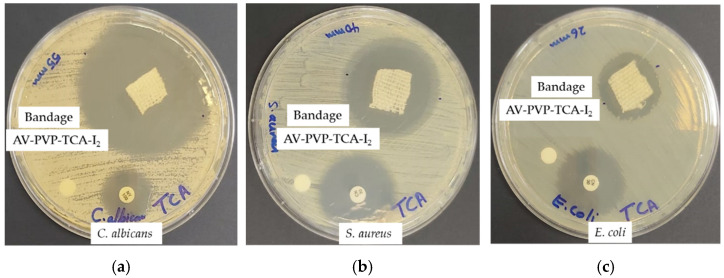
Impregnated sterile bandages with AV-PVP-TCA-I_2_. with positive control antibiotic nystatin (NY = 100 IU) and gentamicin (G = 30 µg/disc). From left to right: AV-PVP-TCA-I_2_ (11 µg/mL) against (**a**) *C. albicans* WDCM 00054; (**b**) *S. aureus* ATCC 25932; (**c**) *E. coli* WDCM 00013.

**Table 1 polymers-14-01932-t001:** Raman shifts in AV-PVP-TCA-I_2_ (1) (cm^−1^).

Group	AV-PVP-TCA-I_2_	[41]	[90]	[91]	[23]	[15]	[92]
I_2_ [I_2_^…^I^−^]	sh,m84* sh,m159*ν_as_	sh,w80*	m85* m160*ν_as_				
	m166*ν_as_ m169*ν_as_ sh,m171*ν_s_ w176*ν_s_	s169*ν_as_			s169 ν_s_ I_2_		
	w179*ν_s_						
	vw183*ν_s_ vw187*ν_s_	w189*ν					
I_3_^−^ [I-I-I^−^]	sh,s62ν_2bend_ s66 δ_def_ sh,m82ν_2bend_ m91ν_1,s_ w100ν_1,s_ vs112ν_1,s_ w238^+^2ν_1,s_	sh,w61 δ_def_ sh,w70 ν_2bend_	sh60 δ_def_ sh,w75ν_2bend_				m85ν_2bend_
		vs110ν_1,s_ vw222^+^2ν_1,s_	s110ν_1,s_ vw221 2ν_1,s_	114ν_1,s_	vs111ν_s_	vs116ν_1,s_ vw235 2ν_1,s_	108ν_s_ 110ν_s_ 218^+^ν_s_
I_3_^−^ [I-I^…^I^−^]	sh,w125ν_3,as_					m126ν_3,as_	
	vw145ν_3,as_	m144ν_3,as_	m144ν_3,as_	144ν_3,as_	m145ν_s_		
	vw152ν_3,as_ sh,m162ν_3,as_		sh,vw154ν_3,as_				159ν_as_
	w325^+^ν_as_	vw334^+^ν_as_					322^+^ν_s_ 434^+^ν_s_ 542^+^ν_s_

ν = vibrational stretching, _s_ = symmetric, _a_ = asymmetric, 1 = stretching mode 1, 3 = stretching mode 3, bend = bending, δ_def_ = deformation. * belong to the same asymmetric, nonlinear unit I_3_^−^ = I_2_^…^I^−^. ^+^ overtones of triiodide ions. vw = very weak, s = strong, vs = very strong, m = intermediate, sh = shoulder.

**Table 2 polymers-14-01932-t002:** UV-vis absorption signals in the samples AV-PVP-TCA (1), AV-PVP-TCA-I_2_ (2), AV-PVP (3), AV-PVP-I_2_ (4), PVP-I_2_ (5) and [58] (nm).

Group	1	2	3	4	5	[58]
I_2_		207 vs 210 vs		205	206 vs	
I_3_^−^		290 s,sh 359 w,br 440 vw,sh		290 358	288 m,br 356 w,br 440 vw,br	

I^−^		202 vs		202	201 vs	
AV/Aloin	208 **	207 **	207 vs			
PVP	203–215 **	203–219 **	201 vs	222	201 vs	
	210 vs	210 vs	202 vs		203 vs	
	216 sh	218 s,sh	209 vs		210 vs,br	
			211 br		215 vs,br	
			213 sh 217 sh		217 vs,br 221 s,sh	
PVP-I_2_	305 w,sh	305 m,sh	305	305	305 w,sh	
TCA	203–215 ** 278 vs 283 s,sh 286 s,sh	203–219 ** 278 vs 283 vs 286 vs				209 218 276

** The broad bands overlap, and several peaks related to AV compounds, iodine moieties and TCA cannot be observed. vw = very weak, br = broad, s = strong, vs = very strong, m = intermediate, sh = shoulder.

**Table 3 polymers-14-01932-t003:** FTIR analysis [cm^–1^] of AV-PVP-TCA (A) and AV-PVP-TCA-I_2_ (B).

	ν_1,2_ (O–H)_s,a_ ν (COOH)_a_	ν (C–H)_a_	ν (CH_2_)_s,a_	ν (C-H)_s_ ν (O-H) *	ν (C=O)_a_ ν (C=C)	δ (C-H)_a_ δ (CH_2_) δ (O-H)	ν (C-C)	ν (C-O) ν (N-H)	ν (C-O)
A	3308	2990	2953	2839	1646 (C=C) TCA 1651 (C=O) PVP 1656 (C=O) TCA	1405 1417 **δ (O-H) TCA** 1450 1465	1314 1343	1290 TCA 1292	1014 1065 1074 1113 1205 1223 1275
B	3308	2989	2953	2839	1632 (C=C) TCA 1639 (C=O) PVP 1649 1655 (C=O) TCA	1405 1417 **δ (O-H) TCA** 1425 1450 1465 1495	1314 1343	1289 TCA 1288	1016 1064 1073 1113 1205 1223 1275

* ν = vibrational stretching, δ = deformation, _s_ = symmetric, _a_ = asymmetric.

**Table 4 polymers-14-01932-t004:** XRD analysis of the samples AV-PVP-TCA-I_2_, AV-PVP-Sage-I_2_ (1), AV-PVP-I_2_ (2) [41], AV-PVP-I_2_-NaI (3) [41] and in previous reports (2Theta^o^).

Group	AV-PVP-TCA-I_2_	1	2	3	[95]	[96]	[13]
I_2_	22.8 m 23.8 m 25.5 m 27 m 30 m 37 vw	-	-	28 m 40 w	25 29 36	-	24.5 s 25 s 28 s 37 w 38 w 43 w
	46 w						46 m
PVP	11 m 20 m	13 s	10 s 19 s,br	11 s,br 20 s,br	-	-	-
TCA	11 m	-	-	-	-	10 vs	-
	15 w					19 w	
	17 s					20 w	
	19 vs					23 s	
	24 m 26 m					25 m 27 w	
						29 m 32 w 34 w	
AV	16 m 22 m	14 s	14 s 21 s,br 22 s,br	-	-	-	-

w = weak, br = broad, s = strong, m = intermediate.

**Table 5 polymers-14-01932-t005:** Antimicrobial testing of antibiotics (A), AV-PVP-TCA-I_2_ by disc dilution studies (1,2,3), suture (S), bandage (B) and mask (M). ZOI (mm) against microbial strains by diffusion assay.

Strain	Antibiotic	A	1 ^+^	2 ^+^	3 ^+^	S	B	M
*S. pneumoniae* ATCC 49619	G	18	14	12	0	3	24	20
*S. aureus* ATCC 25923	G	28	17	13	11	7	40	35
*S. pyogenes* ATCC 19615	G	25	16	12	9	4	24	19
*E. faecalis* ATCC 29212	G	25	15	13	0	3	23	20
*B. subtilis* WDCM 00003	G	21	14	0	0	4	23	16
*P. mirabilis* ATCC 29906	G	30	0	0	0	0	0	0
*P. aeruginosa* WDCM 00026	G	23	13	12	0	2	20	20
*E. coli* WDCM 00013	G	23	13	0	0	0	26	16
*K. pneumoniae* WDCM 00097	G	30	13	10	0	2	28	21
*C. albicans* WDCM 00054	NY	16	46	25	13 *	17	55	54

^+^ Disc diffusion studies (6 mm disc impregnated with 2 mL of 11 µg/mL (1), 2 mL of 5.5 µg/mL (2) and 2 mL of 2.75 µg/mL (3) of AV-PVP-TCA-I_2_. A = G Gentamicin (30 µg/disc). NY (Nystatin) (100 IU). Suture (S), bandage (B) and mask (M) impregnated with 2 mL of 11 µg/mL AV-PVP-TCA-I_2_. Gray shaded area represents Gram-negative bacteria. 0 = Resistant. * Further dilution to 1.38 µg/mL yielded ZOI = 9 mm. No statistically significant differences (*p* > 0.05) between row-based values through Pearson correlation.

## Data Availability

Not applicable.

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
