# Peer review of "Antimicrobial Biomaterial on Sutures, Bandages and Face Masks with Potential for Infection Control"

_polymers, 2022, doi:10.3390/polym14101932_

Round 1

Reviewer 1 Report

Title need to change as per the materials used in this study.

In title, first character of each sentence should be capital as per the journal guidelines.

So many keywords please delete it those are not unnecessary. Maximum 6 is ok.

Last 2 paras in introduction section (Line 114-129) need to make paragraph and just highlight what authors did for this paper. Avoid result and discussion part. Indicate the references in the text accordingly polymer chemistry 2019,10 ,1696-1723, International journal of Biological macromolecules 136 (2019) 661-667 and International journal of Biological Macromolecules 171 (2021) 457-464.

Section 2.3 should have written in the better way and clearly and make into one paragraph.

What is chemistry for the preparation of AV-PVP-TCA and AV-PVP-TCA. Please show it in the main manuscript. Please provide the NMR data and zeta potential data.

All the SEM scale bar need to mark it clearly.

Fig 5, 6, 7, 8, are messy. Need to redraw by scientific software using raw data with high resolution. Figures sublevel a b c etc. marked inside the figures. Mention caption x-axis and y axis unit of all figures in common bracket.

In supplementary section all the figures are messy. Need to redraw by scientific software using raw data with high resolution. All the figures make into one file and submit it to journal accordingly. Mark the figure numbering Fig. S1…and so on. Indicate accordingly in the main text.

Smooth the ftir curve its messy.

In fig 8, x-axis just write as 2-theta (deg.)

Once use abbreviation please make constancy throughout the manuscript. Don’t use again full name. Check all the abbreviation.

Line 491, ‘R’ should be small in “Ray’.

Line 513, it should 3.2.4. Check it.

In fig 9,10, 11 and 12. Please clearly write the samples name inside the images. Typing is preferably. At present optical images have low resolution. Please improve it.

Please reduce discussion section. At present so long. Discuss briefly by citing to support your conclusions.

Conclusions section need to be written quantatively with imp results.

Author Response

Title need to change as per the materials used in this study. Dear Reviewer, the title was changed accordingly.

In title, first character of each sentence should be capital as per the journal guidelines. Thanks. Done We changed each word to capital letter.

So many keywords please delete it those are not unnecessary. Maximum 6 is ok. Dear Reviewer, we deleted 2 keywords from the list, hoping to keep the others. Thanks

Last 2 paras in introduction section (Line 114-129) need to make paragraph and just highlight what authors did for this paper. Avoid result and discussion part. Dear Reviewer, changed by removing the results and discussion part. Now it is one paragraph. Thanks.

 Indicate the references in the text accordingly polymer chemistry 2019,10 ,1696-1723, International journal of Biological macromolecules 136 (2019) 661-667 and International journal of Biological Macromolecules 171 (2021) 457-464. Dear Reviewer, these were added as 80 and 81, along with 82, related to chitosan and ferulic acid grafted polymers on lines 107-108. Thanks a lot for this new perspective.

Section 2.3 should have written in the better way and clearly and make into one paragraph. Yes, thanks. Done accordingly.

What is chemistry for the preparation of AV-PVP-TCA and AV-PVP-TCA. Please show it in the main manuscript. Please provide the NMR data and zeta potential data. Dear Reviewer, we prepared a graphical abstract to satisfy this question. We have here a facile, one-pot-synthesis of the biomaterial, which has PVP as polymer, which incorporates iodine and stabilizes the available plant-biomaterials. Unfortunately, due to time restrictions and difficulties in the outsourcing, we were not able to get NMR and zeta-potential data. The biomaterial is not considered nanosized. Thank you so much and sorry, for this inability to provide these analytical data. We hope, you can understand our problem.

All the SEM scale bar need to mark it clearly. Dear Reviewer, the SEM was provided like the other analytical data from outsourcing. Unfortunately, we cannot obtain better and clearer pictures from them.

Fig 5, 6, 7, 8, are messy. Need to redraw by scientific software using raw data with high resolution. Figures sublevel a b c etc. marked inside the figures. Mention caption x-axis and y axis unit of all figures in common bracket. Dear Reviewer, thank you for indicating this as a problem. Actually, we do not have this scientific software and used instead excel, as we did before in all our publications. Unfortunately, we are not able to get this software in short time during these 10 days. We did the editing and tried to remove the messy parts, added common brackets. Thanks. It looks much better now.

In supplementary section all the figures are messy. Need to redraw by scientific software using raw data with high resolution. All the figures make into one file and submit it to journal accordingly. Mark the figure numbering Fig. S1…and so on. Indicate accordingly in the main text. We edited the pictures without compromising the resolution further. We re-numbered the supplementary figures and removed the double S3. In the manuscript, we used only S3 without description (Raman full w TCA). We also edited the FTIR as much as we could. Thanks

Smooth the ftir curve its messy. Dear Reviewer, we worked on the FTIR under our given possibilities. It was smoothed. We tried to improve it more, but it seems, we reached our limits. We hope, the outcome will be satisfactory. Thank you.

In fig 8, x-axis just write as 2-theta (deg.) Dear Reviewer. This picture was directly provided by the outsourcing institute. Changing the x-axis needed editing the picture by cropping the jpeg, editing and saving again as a new jpeg. This procedure makes the picture unfortunately less clear and useless. We could not do better, so we kept the original picture. Hope, you can understand us. Sorry and Thank you so much. Next time we outsorce again, we will ask for this description as x-axis.

Once use abbreviation please make constancy throughout the manuscript. Don’t use again full name. Check all the abbreviation. Thank you so much dear Reviewer, we did our best to eliminate this mistake.

Line 491, ‘R’ should be small in “Ray’. Done, thanks.

Line 513, it should 3.2.4. Check it. Indeed ! Changed to 3.2.4. Thanks !

In fig 9,10, 11 and 12. Please clearly write the samples name inside the images. Typing is preferably. At present optical images have low resolution. Please improve it. Dear Reviewer, we tried to get better results, but unfortunately we were not able to improve the quality by technical means. We used the same method, as we did in our previous publications. We cannot use the same plates again for new photos, because they are already discarded. Repeating all the antimicrobial testing would mean loss of time and resources at the same time. We are sorry, not being able to deliver sharper pictures. The samples are all the same, as indicated below in the description. It is the sample AV-PVP-TCA-I2. Writing inside the picture made the pictures messy and cover up the ZOI, so unfortunately, we could not follow your advice. To make it complete, I added all the pictures as jpeg.

Please reduce discussion section. At present so long. Discuss briefly by citing to support your conclusions. Actually, you are right dear reviewer. We were feeling also, that it is too long. Therefore, we did our best to reduce, modify and edit the text. For shortening, we combined results and discussion as an effort of major revision. Thanks again.

Conclusions section need to be written quantatively with imp results. We added the lines 966-970.

Thank you so much for your support and valuable comments. We improved our manuscript by following your important feedback. We added the needed references and followed your comments as much as possible. Thank you again for your efforts to improve the manuscript.

Best regards

Zehra

Reviewer 2 Report

Dear author, please revise your manuscript to the following suggested points

I strongly recommend revising this manuscript as follows:

  1. I would like to recommend the authors, to enhance the readers understanding please draw a schematic diagram to understand the whole story of the manuscript which you want to convey to the readers also include the real pictures of the materials and chemical structures in it.

  1. I would like to suggest the authors please refer to the following recent article biomaterials based article, Advancement of Biomaterial‐Based Postoperative Adhesion Barriers." Macromolecular bioscience 21, no. 3 (2021): 2000395.

  1. Authors should add the reported information about a variety of synthetic/natural/ hybrid materials in the introduction part author should be added the examples of the natural and synthetic systems which are having several advantages such as pH responsiveness, enzymatic degradability, and tailoring the physicochemical properties, for that author should cite DOI https://doi.org/10.1039/9781788015769-00047, https://doi.org/10.1021/acsami.5b10675 and DOI: 1039/C5TB01251A
  2. In the introduction paragraph number authors have discussed about the antimicrobial materials I woud like to suggest to authors cite the following recent articles in this section: A. Silver-loaded carboxymethyl cellulose nonwoven sheet with controlled counterions for infected wound healing." Carbohydrate Polymers286 (2022): 119289. Multifunctionalization of Poly (vinylidene fluoride)/reactive copolymer blend membranes for broad-spectrum applications. ACS applied materials & interfaces, 9(3), 3102-3112. C. Self-assembly of partially alkylated dextran-graft-poly [(2-dimethylamino) ethyl methacrylate] copolymer facilitating hydrophobic/hydrophilic drug delivery and improving conetwork hydrogel properties." Biomacromolecules 19, no. 4 (2018): 1142-1153..
  3. In the figure (a) why the size of the sample is different from (b) and (c) please explain

The author has discussed Zone of Inhibition Plate Studies and Disc Diffusion what is the difference between them please clarify.

  1. Inline number 577 author has used the term mask tissues. What is the meaning of it please explain.
  2. The author should provide the information about the coating material ( how much amount per square area was coated over the materials) this information is quite important please include it in the manuscript.

Author Response

  1. I would like to recommend the authors, to enhance the readers understanding please draw a schematic diagram to understand the whole story of the manuscript which you want to convey to the readers also include the real pictures of the materials and chemical structures in it. Dear Reviewer, this is a very good comment, it will help the reader a lot. Therefore, we included graphical abstracts from our artist amie. I hope, it is now easier for the readers. Thanks a lot !
  1. I would like to suggest the authors please refer to the following recent article biomaterials based article, Advancement of Biomaterial‐Based Postoperative Adhesion Barriers." Macromolecular bioscience 21, no. 3 (2021): 2000395.

Added. Thanks. See below.

  1. Authors should add the reported information about a variety of synthetic/natural/ hybrid materials in the introduction part author should be added the examples of the natural and synthetic systems which are having several advantages such as pH responsiveness, enzymatic degradability, and tailoring the physicochemical properties, for that author should cite DOI https://doi.org/10.1039/9781788015769-00047, https://doi.org/10.1021/acsami.5b10675 and DOI: 1039/C5TB01251A

Added. Thanks. See below.

  1. In the introduction paragraph number authors have discussed about the antimicrobial materials I woud like to suggest to authors cite the following recent articles in this section: A. Silver-loaded carboxymethyl cellulose nonwoven sheet with controlled counterions for infected wound healing." Carbohydrate Polymers286 (2022): 119289. Multifunctionalization of Poly (vinylidene fluoride)/reactive copolymer blend membranes for broad-spectrum applications. ACS applied materials & interfaces, 9(3), 3102-3112. C. Self-assembly of partially alkylated dextran-graft-poly [(2-dimethylamino) ethyl methacrylate] copolymer facilitating hydrophobic/hydrophilic drug delivery and improving conetwork hydrogel properties." Biomacromolecules 19, no. 4 (2018): 1142-1153.

Dear Reviewer, we added the needed citations from Chandel et al from 83-87 into the introduction as advised in lines 108-110. We refrained discussing them in detail, in order not to divert the attention. Thank you so much for the interesting references, which really shed light into novel biomaterials for tissue repair processes.

  1. In the figure (a) why the size of the sample is different from (b) and (c) please explain. Dear reviewers, which figure number

The author has discussed Zone of Inhibition Plate Studies and Disc Diffusion what is the difference between them please clarify. Dear reviewer, the Zone of inhibition plate studies are actually the same as the Disc diffusion, when it is done only with discs soaked with the antimicrobial as in 2.6.2. Because we have also bandages, sutures and masks, we wrote these under different headings. If it causes confusion, we can remove the subheadings and just write all as part of 2.6.1.

  1. Inline number 577 author has used the term mask tissues. What is the meaning of it please explain. Dear reviewer, you are right to highlight this issue, it has actually no meaning, therefore, we removed all. It is just confusing and unnecessary. Thank you !
  2. The author should provide the information about the coating material ( how much amount per square area was coated over the materials) this information is quite important please include it in the manuscript. Dear reviewer, indeed, this is important information. We included it in 2.7. as follows:

“The uncoated, sterile, multifilamented surgical PGA sutures of 2.5 cm were impregnated with AV-PVP-TCA-I2 for 18 h into 50 mL of AV-PVP-TCA-I2 solution (11 µg/mL) at RT. The blue sutures became brown-blue and were then dried for 24 h under ambient conditions. The bandages and surgical face masks were cut to square pieces of (5 cm x 5 cm), were also impregnated in 50 mL of our title formulation (11 µg/mL) for 18 hours at RT and dried for 24 h at RT. These dip-coated, dried sutures, bandages and surgical face masks were tested in vitro by ZOI assay against our selection of 10 microbial strains” (lines 243-249).

Dear Reviewer, thank you so much for your efforts and detailed comments, which helped us to articulate our manuscript better. Especially adding a graphical abstract and the really interesting references improved our manuscript a lot.

Best regards

Zehra

Reviewer 3 Report

in my opinion, the work is very interesting but quite long. It could easily be divided into several parts. I suggest linguistic and editorial corrections

Author Response

in my opinion, the work is very interesting but quite long. It could easily be divided into several parts. I suggest linguistic and editorial corrections

Dear Reviewer,

Indeed, you are right in this comment. Actually, we had the same feeling when looking at the manuscript from your point of view. We did our best to reduce the text by making major revision. We changed results to results and discussion, so we can shorten the text. At the same time, discuss relevant results directly. We did our best to eliminate linguistic mistakes and correct any mistake during the major revision of the manuscript.

Thank you so much for your support and valuable comments. We improved our manuscript by following your important feedback.

Best regards

Zehra

Round 2

Reviewer 1 Report

The authors revised the manuscript partially. But the manuscript is still messy.  All the queries authors have not complied but authors have written it is done which is unacceptable. before the final publication of this manuscript, all the problems need to solve. 

Find the specific comments in the PDF file

Author Response

Reviewer 1

  1. Line 207, 2.4.4. Fourier-Transform Infrared Spectroscopy (FTIR), it should be written as 2.4.4. Fourier-Transform Infrared (FTIR) Spectroscopy.

Thanks a lot. DONE ! (line 200)

  1. Still 9 keywords, keep maximum 6 is ok.

DONE :)

  1. This comment not compile in manuscript. Check it. At present description is too much. Last 2 paras in introduction section (Line 114-129) need to make paragraph and just highlight what authors did for this paper. Avoid result and discussion part. Dear Reviewer, changed by removing the results and discussion part. Now it is one paragraph. Thanks”.

DONE :)

Please see lines 123-133. It contains, what we did for this manuscript and we avoided adding results and discussion.

In the description, we have to mention the theoretical background of iodine, which is our specialization. Additionally, we have to speak about all the other ingredients in our biocomposite in order to give the reader an idea, why we used them. The general problem of surgical site infections and COVID, as well as AMR must be stressed as well, because it is the reason for our ongoing investigations. The problem with face mask and waste management is also one part of our aims to offer solutions for the future. Reducing any content from this, could leave the reader with questions. Therefore, we believe in giving a full, comprehensive description to the reader due to the complexity and importance of the problem. We hope, that this conveys heavily the message of the ongoing AMR crisis and like to keep the introduction in this way without shortening it more.

  1. Reference 82 are different than suggested references. Check it seems authors make mistake.

All is fine, there is no mistake. DONE :)

We found this brand-new reference and we are pleased to add it into the references section as reference 82. It is very informative and needed for the manuscript. The other 5 suggested references are already added as 83,84,85,86 and 87. Please check:)

  1. 5. Section 2.3 is not in one paragraph. Check it again.

2.3. Preparation of AV-PVP-TCA and AV-PVP-TCA-I2

The stock solution AV-PVP is prepared by adding 2 mL pure AV gel into 2 mL of a solution of 1 g polyvinylpyrrolidone K-30 (PVP) in 10 mL distilled water under continuous stirring at room temperature (RT). For the preparation of AV-PVP-TCA, first 0.148 g TCA is dissolved in 10 mL ethanol. Then 2 mL of this solution is added to AV-PVP under continuous stirring at RT. After that, iodine solution is obtained by dissolving 0.05 g of iodine in 3 mL ethanol in a covered beaker at RT under stirring. 1 mL of this iodine solution is added to AV-PVP-TCA under continuous stirring at RT for the preparation of AV-PVP-TCA-I2 .

Dear Reviewer, you mean this section 2.3 (lines 167-175) ? We changed it to 1 paragraph. Or you mean something else, please ?

  1. “What is chemistry for the preparation of AV-PVP-TCA and AV-PVP-TCA. Please show it in the main manuscript. Please provide the NMR data and zeta potential data. Dear Reviewer, we prepared a graphical abstract to satisfy this question. We have here a facile, one-pot-synthesis of the biomaterial, which has PVP as polymer, which incorporates iodine and stabilizes the available plant-biomaterials. Unfortunately, due to time restrictions and difficulties in the outsourcing, we were not able to get NMR and zetapotential data. The biomaterial is not considered nanosized. Thank you so much and sorry, for this inability to provide these analytical data. We hope, you can understand our problem.” Reviewer understand authors institute have not available such type of instrument.

Thank you so much for your understanding!

In my previous comment; it is not complied where is your graphical abstract?

We added a graphical abstract and uploaded it separately as graphical abstract. Maybe it went missing. We attached it below as picture.

 I suggest to draw a schematic diagram to show the interaction. Put all the new figures in the cover letter as well as main manuscript.

DONE :)

We asked our artist "@art_by_amie" to prepare a schematic diagram above and attached is as graphical abstract. She likes us to keep these pictures in one in picture as acknowledgement of her work for us.

We will put this into the cover letter of course.

On which place, we may put it into the main manuscript ? It was intended as a graphical abstract in agreement with amie. By this way, it will be seen on the webpage as first picture next to the abstract. In most of our previous publications, we received thankfully graphical abstracts from our artist.

  1. All the SEM scale bar need to mark it clearly. Dear Reviewer, the SEM was provided like the other analytical data from outsourcing. Unfortunately, we cannot obtain better and clearer pictures from them. Here I suggest scale bar to draw by power point by measuring original scale bar.

DONE :)

It looks much nicer now ! Thanks !

  1. Fig 5, 6, 7, 8, are messy. Need to redraw by scientific software using raw data with high resolution. Figures sublevel a b c etc. marked inside the figures. Mention caption x-axis and y axis unit of all figures in common bracket. Dear Reviewer, thank you for indicating this as a problem. Actually, we do not have this scientific software and used instead excel, as we did before in all our publications. Unfortunately, we are not able to get this software in short time during these 10 days for the revision of the manuscript. We did the editing and tried to remove the messy parts, added common brackets. Thanks. It looks much better now. Scientific software (origin or sigma plot) available in online free version. Your all figures are not standard to publish this manuscript in polymers journal. Please do it.

Hopefully considered DONE :)

We removed all the UV-vis and kept only one important picture. If needed, we can even remove this picture and all the other figures and keep it in the supplementary material. We anyhow have provided comprehensive tables within the manuscript.

(Dear Reviewer, we improved again all the figures with the means we have now, which is EXCEL. Please understand our condition. We used EXCEL in all our publications and it was never an issue until now and no one told us about Origin nor Sigma plot. Thank you for your important advice.

For our next publication, surely we will ask IT department to install that free software on our devices. In order to reduce the messy appearance, we removed unnecessary figures. Please consider our inability now, to do the needed for the free software due to ongoing public holidays of 10 days, starting from today. Additionally, we will need to wait for the appointment of IT to do the actual download after the holiday break as well. Thank you for your understanding and support !)

  1. In supplementary section all the figures are messy. Need to redraw by scientific software using raw data with high resolution. All the figures make into one file and submit it to journal accordingly. Mark the figure numbering Fig. S1…and so on. Indicate accordingly in the main text. We edited the pictures without compromising the resolution further. We re-numbered the supplementary figures and removed the double S3. In the manuscript, we used only S3 without description (Raman full w TCA). We also edited the FTIR as much as we could. Thanks. Here make all the supplementary files into one word or PDF file. And same also figure need to redraw.

DONE :)

We put all the edited figures and supplementary figures separately in two new pdf files as requested. Thanks !

  1. Smooth the ftir curve its messy. Dear Reviewer, we worked on the FTIR under our given possibilities. It was smoothed. We tried to improve it more, but it seems, we reached our limits. We hope, the outcome will be satisfactory. Thank you. Smooth you can do using origin or sigma plot software. Its not a big deal but need to so it.

Hopefully considered DONE :)

Because we could not download on our university owned computers the software, we included into the manuscript the original FTIR from the outsourcing institute. We hope, this solves the problem without needing origin. Another suggestion would be, to remove the two FTIR from the manuscript and keep it in the supplementary file because we provided anyhow a comprehensive table of the analysis. Thanks !

(We are sure it is not a big deal to use those kind of software, but, we need admin from university IT department to be able to download any program on our laptops. We have now 10 days holidays and no one will be able to help us from the IT department. Even after the 10 days, we need to raise a request and wait for appointment from IT, to send us someone for downloading the software. Hope, you understand our situation and kindly accept the solution we provided. Thanks.)

  1. 11. In fig 8, x-axis just write as 2-theta (deg.)

DONE :)

It’s not way to present the figure raw data; you have raw data you can draw good figure using software. Please do it.

Dear Reviewer, we did not find the raw data. The outsourcing institute send us the picture only. This was before 1 year, the data is deleted. We edited the picture regarding the x-axis. We do not have sample, nor any appointment to repeat the XRD in near future. Please accept our apology.

However, we added as previously requested the x-axis and this would solve the original problem hopefully. Thanks again.

  1. In fig 9,10, 11 and 12. Please clearly write the samples name inside the images. Typing is preferably. At present optical images have low resolution. Please improve it. Dear Reviewer, we tried to get better results, but unfortunately we were not able to improve the quality by technical means. We used the same method, as we did in our previous publications. We cannot use the same plates again for new photos, because they are already discarded. Repeating all the antimicrobial testing would mean loss of time and resources at the same time. We are sorry, not being able to deliver sharper pictures. The samples are all the same, as indicated below in the description. It is the sample AV-PVPTCA-I2. Writing inside the picture made the pictures messy and cover up the ZOI, so unfortunately, we could not follow your advice. To make it complete, I added all the pictures as jpeg. By using power point, you can write inside the text. Hand written is there no problem. But type the sample name inside the figure using power point.

DONE and it is looking good :) Thanks !

Thank you for your commitment to improve the manuscript. We are happy to have followed your important, valuable advises. We did our best to follow everything you commented. Only, we were not able to use origin for the figures. For this reason, we added two new original FTIR, edited the existing UV and Raman, removed unnecessary pictures and have the suggestion to even move all those pictures into supplementary materials section because we already provided in every section a comprehensive table for the figures. This is our suggestion to solve the problem.

We hope for your understanding and support.

Thank you so much for your valuable efforts

Best regards

Zehra

Round 3

Reviewer 1 Report

The Authors have improved the manuscript.

Author Response

Thank you so much for your kind efforts dear Reviewer !

Indeed, we improved the manuscript by your valuable efforts.

Best regards

Zehra

This manuscript is a resubmission of an earlier submission. The following is a list of the peer review reports and author responses from that submission.

Round 1

Reviewer 1 Report

This study reports very curious and interesting results. Unfortunately, the presentation of the manuscript has many negative points. The experiments conducted in the study do not follow or adhere to any experimental guidelines for new molecules. Many reliable tests currently on the market have not been carried out, moreover, nothing is known about a possible cytotoxic effect, easily performed through an MTT assay. Even the manuscript appears to be not very linear and very confusing, should be simpler and clearer.

Reviewer 2 Report

I think this manuscript really is a merit to the reader.